# Mitigating Semantic Collapse
# in Partially Relevant Video Retrieval

**WonJun Moon**[†]**, MinSeok Jung**[†]**, Gilhan Park, Tae-Young Kim,**
**Cheol-Ho Cho, Woojin Jun, Jae-Pil Heo**[∗]
Sungkyunkwan University
{wjun0830, alstjr88, a01152a, jackdawson,
gersys, junwoojin, jaepilheo}@skku.edu

## Abstract

Partially Relevant Video Retrieval (PRVR) seeks videos where only part of the content matches a text query. Existing methods treat every annotated text–video pair as a positive and all others as negatives, ignoring the rich semantic variation both within a single video and across different videos. Consequently, embeddings of both queries and their corresponding video-clip segments for distinct events within the same video collapse together, while embeddings of semantically similar queries and segments from different videos are driven apart. This limits retrieval performance when videos contain multiple, diverse events. This paper addresses the aforementioned problems, termed as semantic collapse, in both the text and video embedding spaces. We first introduce Text Correlation Preservation Learning, which preserves the semantic relationships encoded by the foundation model across text queries. To address collapse in video embeddings, we propose Cross-Branch Video Alignment (CBVA), a contrastive alignment method that disentangles hierarchical video representations across temporal scales. Subsequently, we introduce order-preserving token merging and adaptive CBVA to enhance alignment by producing video segments that are internally coherent yet mutually distinctive. Extensive experiments on PRVR benchmarks demonstrate that our framework effectively prevents semantic collapse and substantially improves retrieval accuracy.

## 1 Introduction

Recently, Partially Relevant Video Retrieval (PRVR) [6, 47, 46] has emerged as a significant research challenge in computer vision. PRVR shares the same objective as traditional Text-to-Video Retrieval [26, 36, 30, 13, 16, 31], retrieving the video that best aligns with a given text query. However, the key difference lies in PRVR's assumption that target videos may be only partially relevant to the query rather than requiring a perfect semantic match. The primary challenge in PRVR lies in learning from text-video pairwise annotations. A single video is often associated with multiple distinct text queries labeled as positive pairs; however, the semantic relationships among these text queries are not explicitly defined, and fine-grained temporal annotations that indicate their precise alignment within the video are typically unavailable.

As a result, conventional training for retrieval based on the InfoNCE loss [3, 21] induces a semantic collapse problem in PRVR. Semantic collapse refers to the phenomenon where paired text queries and visual segments are excessively attracted to each other while being indiscriminately repelled from features of other pairs, regardless of their actual semantic similarity. Fig. 1 (a) illustrates this issue within the text embedding space; text queries associated with the same video tend to cluster

---

[∗]Corresponding author
[†]Equal contribution.

39th Conference on Neural Information Processing Systems (NeurIPS 2025).

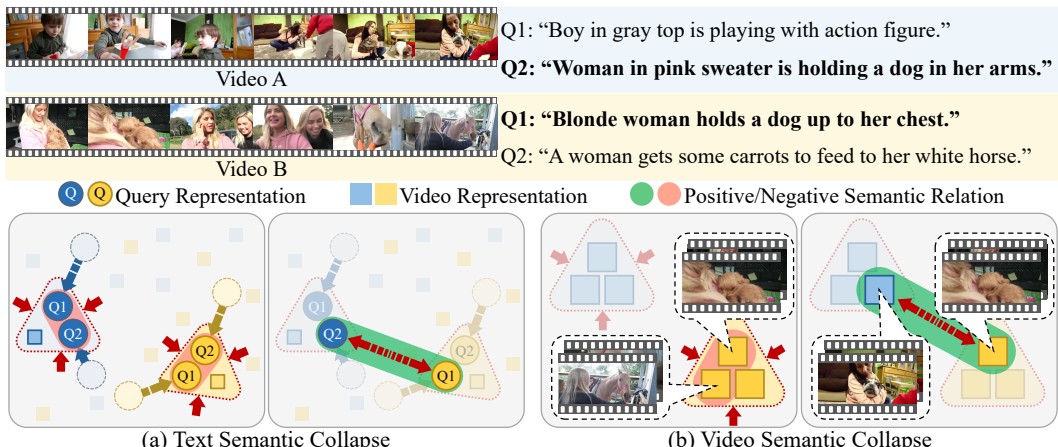

Figure 1: Illustration of semantic collapse. (**Up**) Untrimmed videos in PRVR encompass diverse semantics that can be described by different texts. As a result, semantic segments (both text and video clips) from the same video may convey very different meanings, while segments from different videos can nonetheless be closely related. For example, Q2 of Video A and Q1 of Video B both depict "holding a dog". (**Down**) Since all queries tied to a given video are treated as positives and negative queries drawn from other videos, the model pulls together all text embeddings (and their corresponding video segments) for that video, regardless of true meaning, and pushes apart semantically similar queries (and segments) from different videos. (a) illustrates that queries of the same video are pulled together regardless of their semantic relationships (left), while queries with similar context (holding a dog) are pushed apart (right). (b) shows that video segments also suffer from the same phenomenon.

together even when they are semantically unrelated, while semantically similar queries are pulled apart when they are paired with different videos. In addition, the same phenomenon occurs in video embeddings; video segments drawn from the same video collapse together regardless of their true semantic differences, as shown in Fig. 1 (b). This is because the training guidance is provided by video ID, not by their individual semantic content. In short, every segment in a video shares the identical set of paired text queries as positives.

Previous works, e.g., GMMFormer [47] and GMMFormer-v2 [46], have attempted to address the semantic collapse within text embeddings. Specifically, these methods explicitly reduce the similarity between text queries paired with the same video. However, the semantic relationships between text queries are often overlooked, and the issue of semantic collapse within video embeddings remains underexplored, leading to sub-optimal performance.

In this paper, we aim to mitigate the semantic collapse in both text and video embeddings for PRVR. First, we introduce Text Correlation Preservation Learning (TCPL), which leverages CLIP [37], a vision-language foundation model with a well-structured semantic space. By distilling the semantic relationships encoded in CLIP, TCPL effectively regularizes the semantic collapse within text embeddings. While TCPL leverages CLIP's rich text-semantic structure to regularize collapse in the textual embedding space, we point out that the same approach cannot be directly applied to video embeddings. This is because CLIP's pretraining operates on static images, thereby lacking the capacity to model temporal dynamics [27].

To this end, we introduce Cross-Branch Video Alignment (CBVA), a dedicated objective to preserve context diversity in the video modality. CBVA utilizes a dual-branch architecture commonly adopted in PRVR to encode hierarchical video representations and employs a contrastive objective to differentiate distinct events within a video. Concretely, frame- and clip-level embeddings from the same timestamp are encouraged to align closely, while those from different timestamps are driven apart. Then, we further leverage the token merging strategy in two ways to enhance video-adaptivity within CBVA; (1) order-preserving token merging is introduced for semantically consistent video clip aggregation, and (2) bipartite token merging [1] is leveraged to organize representative contexts within each video. By encoding clips in a context-aware manner, we encourage videos to be represented in line with their true semantic content. Consequently, with TCPL and CBVA combined, our method achieves state-of-the-art performances in all tested benchmarks.

In summary, our contributions are (1) We propose Text Correlation Preservation Learning, which leverages the semantic relationships within the foundation model to address semantic collapse within text embeddings, (2) We propose Cross-Branch Video Alignment to mitigate the semantic collapse in video modality by distinguishing distinct events within a video, (3) We leverage token merging strategies to encourage the precise video alignment, and (4) Our method achieves superior performances across all datasets in PRVR.

## 2    Related Work

**Partially Relevant Video Retrieval.** PRVR aims to retrieve untrimmed videos that are partially relevant to a given query [6, 19, 20, 51]. MS-SL [6] addresses this challenge by proposing a dual encoding strategy that explicitly separates features for frame and clip segments, capturing different temporal scales within untrimmed videos. Subsequently, DL-DKD [7] leverages CLIP [37] to enhance PRVR performance by distilling text–frame similarity. GMMFormer [47] introduces a Gaussian Mixture Model–based Transformer that enables efficient retrieval with a reduced set of video features. It also identifies semantic collapse as a key challenge and proposes a query-diverse loss to enforce separation among multiple text queries linked to the same video. Building on this, GMMFormer v2 [46] further addresses semantic collapse by explicitly controlling the degree of semantic separation between queries associated with the same video. Unlike these methods that only enforce separation among a small set of queries, our approach aims to leverage their true semantic relationships and additionally mitigates semantic collapse in the video embedding space.

**Knowledge Distillation.** The aim of knowledge distillation is to train a student model with fewer parameters to achieve performance comparable to a larger teacher model [15]. For classification tasks, Kullback-Leibler divergence loss is widely applied to align the student's output distribution with that of the teacher after the softmax layer, allowing the student model to learn from the teacher's predictions. Subsequently, transferring knowledge at the intermediate feature level has been the next stream [45, 18, 4]. However, as they fail to effectively capture the relationships between individual features, Relational Knowledge Distillation (RKD) [35, 29, 41] was proposed to distill the relationships within the semantic space of the teacher model to that of the student. In PRVR, the problem of semantic collapse occurs due to the lack of consideration for relationships among queries paired with the same video, as well as across queries from different videos. Therefore, we leverage RKD to transfer structured semantic relationships within the foundational model to typical PRVR network designs [6, 47, 46] that often suffer from semantic collapse.

**Token Merging.** Token merging [1, 2, 34] has been proposed to improve the efficiency of Transformer [42] by reducing token redundancy. A representative method, ToMe [1], uses bipartite matching on token similarities to merge spatial tokens in the vision transformer. Recently, token merging strategies have been extended to the video domain. For example, LearnableVTM [23] learns per-patch saliency scores and applies for merging across long videos. TempMe [38] sequentially merges tokens within progressively larger fixed-window clips, addressing both spatial and temporal redundancy for retrieval. In contrast, our work applies token merging for two purposes: we merge semantically-coherent adjacent video frames to assemble coherent contexts in each video clip, and leverage token merging to determine the representative context within each video. These facilitate precise alignment between hierarchical video representations.

## 3    Method

### 3.1    Preliminary

Our architectural design is illustrated in Fig. 2. Similar to prior works, we employ pretrained encoders to extract tokens, which are processed through trainable layers.

**Text encoder.** Given a batch of text inputs, we utilize the pre-trained text encoder to extract text tokens $T \in \mathbb{R}^{B_q \times L_q \times d_q}$, where $B_q$, $L_q$ and $d_q$ denote the number of text queries, the number of words per query, and the dimension of query representation, respectively. The sequence of word tokens includes [SOS] (start of sequence) at the beginning and [EOS] (end of sequence) at the end, making the total number of tokens $L_q$. These tokens are forwarded through projection layers and transformer layers to produce text representations $\hat{T} \in \mathbb{R}^{B_q \times L_q \times d}$ for downstream text-video retrieval, where $d$ denotes

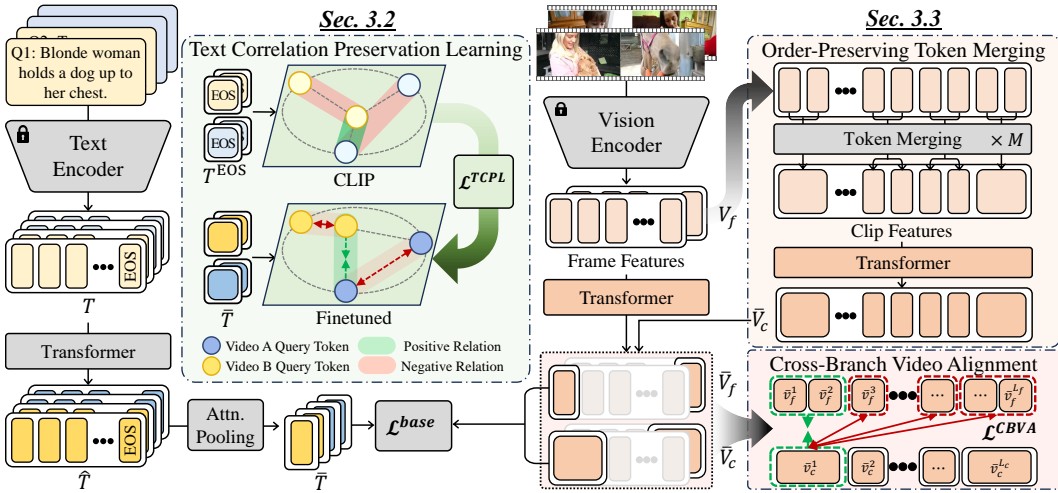

Figure 2: Method overview. We extract text and visual tokens with pretrained backbones, which are then processed via transformer layers. Text tokens are aggregated via attention pooling to produce a single query token $\bar{T}$ for each text query. Also, following prior works, dual-branch visual tokens are encoded (both frame- and clip-level), producing a sequence $\bar{V}$ of video tokens for each level. A baseline retrieval loss $\mathcal{L}^{\text{base}}$ aligns $\bar{T}$ with the most similar video token at each level. To mitigate text-side semantic collapse, Text Correlation Preservation Learning transfers CLIP's query relationships. On the other hand, Cross-Branch Video Alignment aligns hierarchical segments by timestamping to mitigate collapse and preserve visual details. Furthermore, CBVA is precisely enhanced by constructing coherent clips with Order-Preserving Token Merging and improving adaptivity (illustrated in Sec. 3.3).

the projected dimension. Finally, attention pooling is applied to $\hat{T}$ to derive a single aggregated token $\bar{T} \in \mathbb{R}^{B_q \times d}$ that represents the final representation of the text query.

**Video encoder.** For a batch of $B_v$ videos with $L_f$ frames each, we utilize the pre-trained image or video encoder to extract a visual token (e.g. [CLS] token from CLIP) for each frame, generating frame tokens $V_f \in \mathbb{R}^{B_v \times L_f \times d_v}$. Additionally, to represent moments of varying temporal lengths, the frame tokens $V_f$ are aggregated into video clips in the clip branch, to generate clip-level tokens $V_c \in \mathbb{R}^{B_v \times L_c \times d_v}$, where $L_c$ denotes the number of clips per video. Note that our clip construction process is performed with order-preserving token merging, which is discussed in Sec. 3.3. Then, each frame and clip token is encoded independently through the transformer layers to capture contextual relationships. Consequently, $\bar{V}_f \in \mathbb{R}^{B_v \times L_f \times d}$ and $\bar{V}_c \in \mathbb{R}^{B_v \times L_c \times d}$ are produced for final video representations.

**Training objective.** To retrieve a video with the given text query, we perform similarity matching between the representations from two modalities. Specifically, during training, we first select one video token per video that yields the highest similarity to the given text query in both frame and clip branches. Then, these video tokens (one from each video representation) are used to conduct retrieval for training using InfoNCE loss [3, 21] and triplet ranking loss [8]. Accordingly, the final training objective is formulated as follows.

$$\mathcal{L}^{\text{base}} = \mathcal{L}_c^{\text{nce}} + \mathcal{L}_c^{\text{trip}} + \mathcal{L}_f^{\text{nce}} + \mathcal{L}_f^{\text{trip}}, \tag{1}$$

where $\mathcal{L}_*^{\text{nce}}$ and $\mathcal{L}_*^{\text{trip}}$ indicate the InfoNCE loss and triplet ranking loss, respectively, and $\mathcal{L}_c^*$ and $\mathcal{L}_f^*$ represent the clip-level loss and frame-level loss, respectively.

**Problem definition: semantic collapse.** Existing PRVR approaches suffer from semantic collapse which indicates that the general relationships among queries and videos are disrupted. This phenomenon occurs because pairwise text-video annotations (which only specify positive relationships) are used for learning PRVR. Specifically, in PRVR, each video is associated with multiple distinct text queries, which triggers the typical contrastive learning to encourage the queries paired with the same video to cluster together, while text queries paired with different videos are separated as they are attracted to different videos. In this work, we attempt to alleviate the semantic collapse within the text embedding in Sec. 3.2 and video embedding in Sec. 3.3.

## 3.2 Semantic Collapse in Text Embeddings: Text Correlation Preservation Learning

Previously, GMMFormer [47] and GMMFormer-v2 [46] have attempted to address semantic collapse in that they enforced separation between text queries paired with the same video. However, we argue that they only partially alleviate the semantic collapse since all text queries paired with the same video are pushed apart without considering their actual semantic relationship.

To mitigate this issue, we propose Text Correlation Preservation Learning (TCPL), which leverages the well-structured semantic space of CLIP. Specifically, TCPL explores the semantic relationships between text queries within the CLIP semantic space and distills the relationships toward the retrieval space. In this work, we measure the relationships with two metrics: Euclidean distance and angular distance. These two metrics are defined with the pair $(\mathbf{x}, \mathbf{y})$ and triplet $(\mathbf{x}, \mathbf{y}, \mathbf{z})$, where $\mathbf{x}, \mathbf{y}$, and $\mathbf{z}$ denote text embeddings, respectively, as follows:

$$f^{\mathrm{e}}(\mathbf{x}, \mathbf{y}) = \frac{1}{\mu} \|\mathbf{x} - \mathbf{y}\|_2 \; ; \; f^{\mathrm{a}}(\mathbf{x}, \mathbf{y}, \mathbf{z}) = \left\langle \frac{\mathbf{x} - \mathbf{y}}{\|\mathbf{x} - \mathbf{y}\|_2}, \frac{\mathbf{z} - \mathbf{y}}{\|\mathbf{z} - \mathbf{y}\|_2} \right\rangle. \tag{2}$$

$f^e$ and $f^a$ denote Euclidean and angular distance functions, respectively. $\mu$ represents the average distance among all tokens in the mini-batch and $\langle \mathbf{x}, \mathbf{y} \rangle$ denotes the dot product of $\mathbf{x}$ and $\mathbf{y}$.

To measure the semantic relationships within the text embedding space of CLIP, we first gather [EOS] tokens of CLIP in the mini-batch. We define the set of [EOS] tokens in a mini-batch as follows:

$$T^{\mathrm{EOS}} = \{T_{1,L_q}, T_{2,L_q}, \ldots, T_{B_q,L_q}\} \in \mathbb{R}^{B_q \times d_{\mathrm{CLIP}}}, \tag{3}$$

where $T_{1,L_q}$ represents the [EOS] token of the first text query within the mini-batch. Note that [EOS] is used for the distillation since [EOS] conveys more informative clues than other tokens in CLIP [49] and using [EOS] reduces computational overhead compared to token-wise distillation. Then, the knowledge of CLIP is distilled towards the encoded text tokens, $\bar{T}$. Specifically, we distill the pairwise Euclidean distance relationships and triplet angular distance relationships from the CLIP text embeddings into the text-video joint embedding space. The distillation process is expressed as:

$$\mathcal{L}^{\mathrm{E}} = \frac{1}{B_q(B_q - 1)} \sum_{\substack{i,j \in \mathcal{B}_q \\ i \neq j}} \mathcal{L}^{\mathrm{H}}\big(f^{\mathrm{e}}(T_i^{\mathrm{EOS}}, T_j^{\mathrm{EOS}}), f^{\mathrm{e}}(\bar{T}_i, \bar{T}_j)\big), \tag{4}$$

$$\mathcal{L}^{\mathrm{A}} = \frac{1}{B_q^{\;3}} \sum_{i,j,k \in \mathcal{B}_q} \mathcal{L}^{\mathrm{H}}\big(f^{\mathrm{a}}(T_i^{\mathrm{EOS}}, T_j^{\mathrm{EOS}}, T_k^{\mathrm{EOS}}), f^{\mathrm{a}}(\bar{T}_i, \bar{T}_j, \bar{T}_k)\big), \tag{5}$$

where $\mathcal{B}_q = \{1, 2, \ldots, B_q\}$ stands for a set of indices such that $|\mathcal{B}_q| = B_q$ and $\mathcal{L}^{\mathrm{H}}$ denotes Huber loss [14], which leads stable training by behaving as L2 loss for small errors and L1 loss for large errors. Finally, the objective for TCPL is defined as follows:

$$\mathcal{L}^{\mathrm{TCPL}} = \lambda^E \mathcal{L}^{\mathrm{E}} + \lambda^A \mathcal{L}^{\mathrm{A}}, \tag{6}$$

where $\lambda^E$ and $\lambda^A$ are weights for $\mathcal{L}^{\mathrm{E}}$ and $\mathcal{L}^{\mathrm{A}}$, respectively. By preserving the well-structured semantic relationships within the foundation model, TCPL mitigates semantic collapse within text embeddings.

## 3.3 Semantic Collapse in Video Embeddings: Cross-Branch Video Alignment

Semantic collapse also occurs within the video modality. While the conventional text-video retrieval loss effectively pushes apart videos with different semantics, it does not explicitly preserve the multi-contextual nature of events within a single video. As a result, contextually distinct segments within the same video may collapse into similar embeddings, limiting intra-video discriminability.

Therefore, we introduce Cross-Branch Video Alignment (CBVA) that aims to disentangle the representations of distinct events within a video, thereby mitigating semantic collapse. Specifically, we leverage the representations from the typical dual-branch architecture used in PRVR frameworks, with separate encoders for clip- and frame-level branches [6, 47]. In CBVA, timestamp correspondence is leveraged to align each video frame with its matching clip segment while repelling it from segments at other timestamps. However, simply aligning different levels of video representation proves ineffective. This issue stems from the common practice of generating clip segments by uniformly average-pooling fixed-length segments [6, 46], which causes each clip to cover multiple contexts that can overlap across adjacent segments.

**Order-Preserving Token Merging.** To address the fragmentation of temporally adjacent content in untrimmed videos, we first introduce Order-Preserving Token Merging (OP-ToMe) to construct consistent clip segments $V_c$, as shown in Fig. 2. Unlike general token-merging schemes that may fuse tokens from arbitrary spatial or temporal locations [1, 38], OP-ToMe restricts all merging operations to pairs of tokens drawn from successive frames, thereby preserving the original playback order (for stable temporal modeling). Concretely, given a sequence of per-frame tokens, we first compute cosine similarities between disjoint adjacent-frame pairs. We then select the approximately top-$N\%$ of most similar adjacent-frame pairs and merge each into a single clip token. This merging procedure is repeated for $M$ iterations until the frames are aggregated into the standard 32 clips used in prior work. At each merge, the two tokens are fused via a size-weighted average of their feature vectors. Note that the proportional attention mechanism [1] is integrated in our framework to account for each token's size (the number of raw frames it represents). By repeating this process, OP-ToMe produces a condensed sequence of clip segments that (1) maintain strict temporal order, (2) retain coherent contextual semantics, and (3) reduce redundant information across frames—properties that are crucial for robust performance in PRVR. We provide the algorithm for OP-ToMe in the Appendix.

**Cross-Branch Video Alignment.** Once the context-consistent clips are constructed via OP-ToMe, we perform cross-branch contrastive learning to encourage fine-grained temporal discriminability within each video. Specifically, each clip token and its corresponding frame tokens are treated as positive pairs, while frame tokens from other temporal moments in the same video are regarded as negatives. This facilitates the model in learning to distinguish between different contextual segments of a single video. Formally, given that $\bar{V}_c = \{\bar{v}_c^{(i)}\}_{i=1}^{L_c}$ and $\bar{V}_f = \{\bar{v}_f^{(j)}\}_{j=1}^{L_f}$ denote the clip-level and frame-level video tokens respectively, we also define the set of associated frames of each clip $i$ as:

$$\mathbb{F}_i = \{\bar{v}_f^j | \delta(j) = i\}, \quad X_i = |\mathbb{F}_i|, \tag{7}$$

where $\delta(\cdot)$ returns the clip index of a frame among the $L_c$ clips. Then, the objective of CBVA is formulated with frame-to-clip and clip-to-frame NCE as:

$$\mathcal{L}^{\text{CBVA}} = -\frac{1}{L_f} \sum_{i=1}^{L_f} \log \frac{\exp(\text{sim}(\bar{v}_f^i, v_c^{\delta(i)}))}{\sum_{j=1}^{L_c} \exp(\text{sim}(\bar{v}_f^i, \bar{v}_c^j))} - \frac{1}{L_c} \sum_{i=1}^{L_c} \log \frac{\sum_{x=1}^{X_i} \exp(\text{sim}(\mathbb{F}_i[x], \bar{v}_c^i))}{\sum_{j=1}^{L_f} \exp(\text{sim}(\bar{v}_f^j, \bar{v}_c^i))}), \tag{8}$$

where $\text{sim}(\cdot, \cdot)$ denotes cosine similarity and $\mathbb{F}_i[x]$ is the $x$-th frame token in the set $\mathbb{F}_i$.

**Adaptive CBVA.** Although CBVA disentangles different contexts within a single video, real-world footage often contains an unknown (potentially variable) number of distinct contexts. Consequently, applying the contrastive objective in Eq. 8 with a fixed clip length $L_c$ may introduce noise: for example, an interview video composed of largely homogeneous frames will nonetheless be split into $L_c$ segments, unnecessarily fragmenting coherent content. To address this, we first estimate the number of contexts in each video and then adaptively aggregate $L_c^*$ representative clips to guide precise CBVA. We employ bipartite token merging [1] to extract representative clip segments, since semantically similar content may occur intermittently or across non-contiguous intervals within a video. However, optimizing the number of semantics per video is costly during the token merging process. Therefore, we instead pre-define a discrete set of clip numbers based on a fixed merge rate, and then match each video to the level that best reflects its internal similarity structure (number of different semantics). To initially establish a discrete set of clip levels, we define $N\%$ to denote the merge rate and $C_{\min}$ to represent the minimum number of semantically different clips in each video. Then, we generate $K$ levels of clip number candidates $\{L_c^i\}_{i=1}^K$ by recording clip number after each merge step as:

$$L_c^1 = L_c, \quad L_c^{i+1} = \max\big(2 \times \lfloor \frac{L_c^i - (L_c^i/2) \times (N/100) + 1}{2} \rfloor, C_{\min}\big), \tag{9}$$

and let $K$ be the largest index for which $L_c^K \geq C_{\min}$. Next, we compute a high-similarity ratio $\omega$ for each video by measuring the fraction of clip-pair cosine similarities (using frozen features from the backbone $V_c$) that exceed a threshold $\tau$. A low $\omega$ indicates many distinct contexts, so we retain the full original clip set ($L_c^* = L_c$). Otherwise, we select the smallest $k \in \{1, \ldots, K\}$ satisfying $\omega > \frac{K-k}{K}$, and perform $k-1$ iterations of bipartite merging at rate $N\%$, yielding $L_c^* = L_c^k$ final clips. We remark that, for simplicity, we use the same merge rate $N\%$ as OP-ToMe. Consequently, in Eq. 8, the original clip segments are replaced with these merged clips to further enhance video adaptivity. Detailed algorithm for both merging processes are provided in the Appendix.

Table 1: Ablation study on QVHighlights dataset.

|  | Model | R1 | R5 | R10 | R100 | SumR |
|---|---|---|---|---|---|---|
| (a) | Baseline | 21.8 | 48.1 | 60.6 | 95.0 | 225.5 |
| (b) | + TCPL | 22.8 | 49.5 | 63.3 | 95.0 | 230.6 |
| (c) | + Naïve CBVA | 22.8 | 49.4 | 63.7 | 95.0 | 231.0 |
| (d) | + OP-ToMe | 24.2 | 50.4 | 63.0 | 94.9 | 232.5 |
| (e) | + Adaptive CBVA | 23.9 | 51.5 | 63.7 | 95.5 | 234.6 |

Table 2: Performance when using variants of video correlation preservation learning instead of Cross-Branch Video Alignment.

| Method | R1 | R5 | R10 | R100 | SumR |
|---|---|---|---|---|---|
| (a) TCPL baseline | 22.8 | 49.5 | 63.3 | 95.0 | 230.6 |
| (a)+ Retrieved segment | 23.4 | 50.4 | 63.4 | 94.6 | 231.7 |
| (a)+ Uniform Sampling | 22.5 | 50.8 | 64.1 | 94.9 | 232.3 |
| Ours | 23.9 | 51.5 | 63.7 | 95.5 | 234.6 |

## 3.4 Total Training Objective

Finally, our total objective with retrieval, TCPL, and CBVA losses is expressed as:

$$\mathcal{L}^{\text{overall}} = \mathcal{L}^{\text{base}} + \mathcal{L}^{\text{TCPL}} + \lambda^{\text{CBVA}}\mathcal{L}^{\text{CBVA}}. \tag{10}$$

# 4 Experiments

**Datasets & Metrics.** We evaluated our method on four PRVR datasets: QVHighlights [24], TVR [25], ActivityNet Captions [22], and Charades-STA [12]. QVHighlights[24] is a collection of news and vlog-style videos, recently reorganized for PRVR[32]. Each video is paired with an average of 3.3 text queries describing semantically diverse segments. TVR [25] is built from scenes across six popular TV shows, with each video annotated by five text queries targeting different segments. The training set contains 17,435 videos and 87,175 queries, while the evaluation set includes 2,179 videos and 10,895 queries. ActivityNet Captions [22] is sourced from YouTube videos, with an average of 3.7 text queries per video. The dataset includes 10,009 videos for training and 4,917 for evaluation. Charades-STA [12] extends the original Charades dataset by adding sentence-level annotations for specific temporal segments. It consists of 13,898 video-sentence pairs for training and 4,233 for evaluation. For evaluation, we use recall-based metrics, which are commonly used in retrieval tasks [43, 11, 48, 17, 9, 44]. We denote this metric as R@$Q$, where $Q$ represents the proportion of queries for which the correct video appears within the top-$Q$ ranked results. Additionally, SumR is the sum of all R@$Q$ used for evaluation, assessing the overall retrieval performance.

**Implementation Details.** For feature extraction, we follow recent works [5, 33, 32]; we extract video features with CLIP-B/32 [37] and Slowfast [10], and use CLIP-B for text embeddings for QVHighlights, and use CLIP-L [37] for encoding both modalities in other datasets. Hyperparameter configurations are adopted from GMMFormer-v2 [46] (e.g., learning rate, batch size, epochs, and optimizer settings) except for the fusing ratio between the frame and clip branches. We assign a frame score weight of 0.6 and a clip score weight of 0.4. All loss coefficients are fixed across datasets: $\lambda^E = 15$, $\lambda^A = 30$, and $\lambda^{\text{CBVA}} = 0.1$. To construct consistent clips with OP-ToMe, we set $N$ to 75% (Note that $M$ is then computed automatically from $N$ to match the number of clips used in prior works [46, 6].) Finally, we set the minimum clip count per video to $C_{\min} = 5$, and set a similarity threshold $\tau$ to 0.7 for QVHighlights, 0.8 for TVR and ActivityNet-Captions, and 0.85 for Charades. The reason behind using varying $\tau$ is that the internal segment-to-segment similarity distributions differ; QVHighlights exhibits the lowest similarities, TVR and ActivityNet-Captions are intermediate, and Charades shows the highest. All experiments are conducted on a single RTX A6000 GPU and an Intel Xeon Gold 6338 CPU (2.00GHz) for all datasets.

## 4.1 Ablation Study

Studies are conducted on QVHighlights, which includes numerous events in each untrimmed video. The default configuration used to generate the reported results is highlighted in grey.

**Component ablation.** To quantify the contribution of each module, we report a component-wise ablation in Tab. 1. Our baseline is built upon GMMFormer-v2 architecture [46], only trained with the standard retrieval loss $\mathcal{L}^{\text{base}}$. Then, we sequentially add Text Correlation Preservation learning (TCPL) and Cross-Branch Video Alignment (CBVA), which are introduced in Sec. 3.2 and Sec. 3.3. Initially, in row (b), incorporating TCPL mitigates semantic collapse in the text embedding space, yielding a notable gain over the baseline. From row (c) to (e), we subdivide the CBVA into (c) Naïve CBVA, (d) adding OP-ToMe, and (e) applying adaptive CBVA. Specifically, the basic CBVA objective

Table 3: Ablation studies of various components on QVHighlights. 'Coef' denotes coefficient.

(a) TCPL ratio.

| $\lambda^E : \lambda^A$ | SumR |
|---|---|
| 1:1 (15,15) | 229.7 |
| 2:1 (30,15) | 231.8 |
| 1:2 (15,30) | 234.6 |

(b) TCPL coef.

| $\lambda^E$ | $\lambda^A$ | SumR |
|---|---|---|
| 5 | 10 | 231.5 |
| 10 | 20 | 233.5 |
| 15 | 30 | 234.6 |
| 20 | 40 | 232.5 |

(c) TCPL Source.

| Model | SumR |
|---|---|
| CLIP-B | 234.6 |
| CLIP-L | 235.6 |
| OpenCLIP-B | 235.4 |
| OpenCLIP-L | 236.4 |

(d) CBVA coef.

| $\lambda^{CBVA}$ | SumR |
|---|---|
| 0.1 | 234.6 |
| 0.15 | 234.9 |
| 0.2 | 232.9 |

(e) Merge rate.

| $N\%$ | SumR |
|---|---|
| 50 | 232.6 |
| 75 | 234.6 |

(f) Threshold $\tau$.

| $\tau$ | SumR |
|---|---|
| 0.5 | 234.3 |
| 0.6 | 233.5 |
| 0.7 | 234.6 |
| 0.8 | 232.6 |

produces only a marginal increase in performance since fixed-length clip segments may encompass multiple overlapping contexts. However, we find that augmenting CBVA with OP-ToMe to construct semantically consistent clip segments drives a performance boost by reducing spurious alignments across events. Finally, dynamically adjusting each video's clip count according to the estimated number of video contexts further refines the alignment, producing a substantial gain. These results confirm that addressing both the text- and video-side semantic collapse is significant for PRVR.

**Video Correlation Preservation Learning (VCPL).** Similar to TCPL, one can assume that we can apply the identical approach to video embeddings to mitigate semantic collapse. However, this direct adaptation is suboptimal since CLIP's video embeddings cannot model temporal dynamics. To substantiate this, Tab. 2 compares VCPL against our CBVA. 'Retrieved segment' is conducted similarly to TCPL; we first select the representative video token for every text query by identifying the token with the highest similarity within the paired videos (using ground-truth pair) and distill the relationships between representative video segments. Also, we study the variant of VCPL where we uniformly sample 4 segments per video and conduct relation learning between all sampled segments from the mini-batch. Although these approaches yield a modest improvement, we find that these variants lag behind CBVA by 2.3 points in SumR. VCPL is applied to both clip and frame branches.

**Loss coefficients.** For our training objective, we control the TCPL loss with $\lambda^E$ and $\lambda^A$, and the CBVA loss with $\lambda^{CBVA}$. In Tab. 3a, we first studied the $\lambda^E : \lambda^A$ over $\{1 : 1, 2 : 1, 1 : 2\}$. Then, in Tab. 3b with a 1:2 ratio, which yields the best performance, increasing both weights to (15, 30) improved performance; beyond that, gains plateaued. For CBVA, in Tab. 3d, performance rose as $\lambda^{CBVA}$ increased up to 0.15, but for simplicity across datasets, we fixed it at 0.1.

**TCPL source model.** By default, we use the pretrained text encoder as the source model for TCPL to provide semantic relationships (CLIP-B for QVHighlights and CLIP-L for other datasets). To assess sensitivity to the source model, we replaced CLIP-B with alternative vision–language encoders and measured SumR on the QVHighlights dataset in Tab. 3c. As observed, swapping in the larger models (e.g., CLIP-L and OpenCLIP-L) increased SumR by up to 1.8 points. These results indicate that TCPL's effectiveness scales with the quality of the source model's semantic structure.

**Token-Merging Ratio.** We use a single merge rate $N\%$ for both OP-ToMe and adaptive CBVA. Empirically, setting $N$ to approximately 75% reduces 128 frames to 32 clips in only a few steps (matching the standard PRVR frame/clip counts), while keeping computational overhead minimal. As Tab. 3e shows, increasing the number of merge iterations while lowering the per-step ratio to 50% actually degraded accuracy. Thus, we fix $N = 75\%$ across all datasets.

**Adaptively measuring video context number.** We determine the optimal number of contexts for each video by thresholding the pairwise similarity among its clips at a value $\tau$. In this work, we vary $\tau$ to evaluate how sensitive our context-count estimation is to this threshold. As shown in Tab. 3f, the adaptive CBVA method exhibits only minor fluctuations across different $\tau$ values, indicating that it is robust to the choice of similarity threshold between 0.5 and 0.8.

### 4.2 Comparison with the State-of-the-Art

**QVHighlights.** In Tab. 4, we report results on QVHighlights [24], a recently introduced benchmark for PRVR. To illustrate, our method outperforms the previous state of the art by up to 8 points in SumR. We attribute these gains to our method's capability to mitigate semantic collapse, especially when videos exhibit frequent and rapid event transitions.

**TVR** & **ActivityNet-Captions** & **Charades.** Tab. 5 reports results on these three datasets. Specifically,

Table 4: Results on QVHighlights. † denotes reproduced results.

| Methods | R1 | R5 | R10 | R100 | SumR |
|---|---|---|---|---|---|
| MS-SL [6] | 20.4 | 46.7 | 60.7 | 94.6 | 222.5 |
| GMMF [47] | 18.2 | 43.7 | 56.7 | 92.5 | 211.1 |
| AMDNet [39] | 17.4 | 40.8 | 55.0 | 93.4 | 206.6 |
| BGMNet [50] | 20.6 | 46.3 | 58.8 | 94.0 | 219.7 |
| GMMF-v2 [46]† | 21.7 | 48.0 | 60.5 | 95.0 | 225.2 |
| ProtoPRVR [32] | 22.6 | 48.8 | 61.3 | 93.9 | 226.6 |
| **Ours** | **23.9** | **51.5** | **63.7** | **95.5** | **234.6** |

Table 5: Performances on TVR, ActivityNet Captions, and Charades-STA using CLIP-L/14 backbone. † are reproduced results, and all results on Charades are reproduced with official codes.

| Method | TVR | | | | | ActivityNet Captions | | | | | Charades-STA | | | | |
|---|---|---|---|---|---|---|---|---|---|---|---|---|---|---|---|
| | R1 | R5 | R10 | R100 | SumR | R1 | R5 | R10 | R100 | SumR | R1 | R5 | R10 | R100 | SumR |
| CLIP zero-shot | 16.2 | 33.5 | 41.8 | 75.7 | 167.2 | 15.1 | 33.9 | 45.1 | 78.9 | 172.9 | 2.0 | 8.1 | 13.6 | 49.4 | 73.1 |
| MS-SL [6] | 31.9 | 57.6 | 67.7 | 93.8 | 251.0 | 14.7 | 37.1 | 50.4 | 84.6 | 186.7 | **3.4** | 11.5 | 18.7 | 62.5 | 96.0 |
| GMMF [47] | 29.8 | 54.2 | 64.6 | 92.5 | 241.1 | 15.2 | 37.7 | 50.5 | 83.7 | 187.1 | 2.7 | 10.5 | 16.7 | 59.4 | 89.3 |
| AMDNet [39] | 27.7 | 52.3 | 63.3 | 92.3 | 235.6 | 14.0 | 36.3 | 49.9 | 84.2 | 184.5 | 2.1 | 7.8 | 13.9 | 57.2 | 81.1 |
| BGM-Net [50] | 31.1 | 56.3 | 66.5 | 93.8 | 247.7 | 15.6 | 37.9 | 51.3 | 85.4 | 190.3 | 3.0 | 11.8 | 18.2 | 63.7 | 96.7 |
| GMMF-v2 [46]† | 34.0 | 59.7 | 69.8 | 94.5 | 258.1 | 17.1 | 40.6 | 53.7 | 85.5 | 196.9 | 3.1 | 11.6 | 18.2 | 61.4 | 94.2 |
| ProtoPRVR [32] | 34.7 | 60.0 | 70.1 | 94.4 | 259.2 | 16.0 | 38.8 | 52.4 | 85.1 | 192.3 | - | - | - | - | - |
| ARL [5] | 34.6 | 60.4 | 70.7 | 94.4 | 260.1 | 15.3 | 38.4 | 51.5 | 85.2 | 190.4 | - | - | - | - | - |
| **Ours** | **35.1** | **61.6** | **71.5** | **94.9** | **263.1** | **17.7** | **42.0** | **55.6** | **86.8** | **202.1** | 3.2 | **12.6** | **20.1** | **63.8** | **99.7** |

Table 6: Inference time (ms) and memory (MB) across varying size of video database.

| Method | Metric | Number of Videos | | | | |
|---|---|---|---|---|---|---|
| | | 100 | 200 | 300 | 400 | 474 |
| MSSL | Time (ms) | 3.09 | 3.85 | 4.66 | 5.14 | 5.58 |
| | Memory (MB) | 717.47 | 796.15 | 874.83 | 954.14 | 1010.89 |
| GMMF | Time (ms) | 1.97 | 1.98 | 1.99 | 2.02 | 2.05 |
| | Memory (MB) | 243.11 | 248.95 | 254.78 | 260.62 | 264.10 |
| GMMF-v2 | Time (ms) | 2.31 | 2.38 | 2.40 | 2.61 | 2.78 |
| | Memory (MB) | 419.75 | 440.18 | 459.62 | 480.55 | 493.46 |
| Ours | Time (ms) | 2.32 | 2.37 | 2.40 | 2.60 | 2.70 |
| | Memory (MB) | 419.75 | 440.18 | 459.62 | 480.55 | 493.46 |

our method achieves state-of-the-art results on all datasets. The performance gains on these datasets are relatively modest compared to QVHighlights, primarily because QVHighlights exhibits very little overlap between different queries and video segments for the same video, making it especially susceptible to semantic collapse. Despite this, our method maintains state-of-the-art performance across all benchmarks, underscoring its generalizability and effectiveness.

**Efficiency.** In Tab. 6, 7, we report inference/training time and memory, along with model parameters and FLOPs on QVHighlights. Reported times are averaged over 5 runs. For the inference, we measure the inference time and memory across database sizes from 100 to 474 videos. As shown, our method attains the second-lowest inference latency

Table 7: Training efficiency and model complexity.

| Training details | MSSL | GMMF | GMMF-v2 | Ours |
|---|---|---|---|---|
| Time / epoch (ms) | 10,934 | 12,828 | 17,223 | 62,641 |
| Memory (MB) | 2,375 | 3,333 | 7,826 | 9,755 |
| Model params (M) | 4.57 | 12.72 | 32.14 | 32.14 |
| FLOPs (G) | 0.37 | 0.99 | 2.78 | 2.78 |

and memory footprint while achieving substantially higher retrieval accuracy. Note that inference time refers to query time since video features are precomputed and cached in practical deployments. Training statistics in Tab. 7 show higher time and memory due to learning fine-grained video context, but this cost is paid offline, whereas inference efficiency governs real-world deployment where latency and memory are critical.

## 4.3 Analysis

Table 8: Semantic similarity comparison between text and video instances per video. *Intra Sim* is the average similarity among instances of the same video, *Total Sim* is the average pairwise similarity across all instances, and *Diff. Norm* is computed as $(\text{Intra Sim} - \text{Total Sim})/(\text{Intra Sim} + \text{Total Sim})$ to represent the normalized gap between Intra Sim and Total Sim.

| Method | Modality | Intra Sim | Total Sim | Diff. Norm | Modality | Intra Sim | Total Sim | Diff. Norm |
|---|---|---|---|---|---|---|---|---|
| GMMF [47] | | 0.1175 | 0.0113 | 0.8245 | | 0.6419 | 0.0623 | 0.8230 |
| GMMF-v2 [46] | Text | 0.1646 | 0.0196 | 0.7872 | Video | 0.6041 | 0.0387 | 0.8796 |
| Ours | | 0.2198 | 0.0813 | 0.4600 | | 0.5531 | 0.0812 | 0.7440 |

**Similarity Structure.** We compare the pairwise similarity between queries (video segments) associated with the same video (*Intra Sim*) and between all instances across videos (*Total Sim*). If the relationship between contexts and their descriptive queries within each video were indistinguishable

from that observed across different videos, *Diff. Norm* would equal 0; if every context within a video were identical, *Diff. Norm* would equal 1. For the analysis, we leverage QVHighlight to assess semantic collapse via similarity structure, as it exhibits relatively minimal semantic overlap among queries within the same video. As shown in Tab. 8, our method substantially reduces *Diff. Norm* to a point where we claim that our method preserves an appropriate level of relative coherence within each video (not too low) while also mitigating semantic collapse (not too high).

**Spearman rank correlation with CLIP.** We assess whether our method effectively preserves the semantic structure compared to baseline approaches. Specifically, we measure how each method preserves the semantic structure of CLIP using Spearman's rank correlation [40]. For the evaluation, we use the pooled text tokens $\bar{T}$ from each PRVR model to compare with the [EOS] tokens within CLIP query embeddings. Tab. 9 demonstrates how our proposed method well preserves the semantic relationships between text queries, thereby mitigating semantic collapse.

Table 9: Spearman's rank correlation with CLIP.

| Method | CLIP |
|---|---|
| Baseline | 35.40 |
| MS-SL [6] | 37.17 |
| GMMF [47] | 36.06 |
| GMMF-v2 [46] | 35.74 |
| Ours | **68.18** |

**Qualitative results.** Fig. 3 shows qualitative retrieval results for a text query. Our method correctly retrieves and localizes the video token that overlaps the query's target moment (within additional temporal margin [52, 28, 32]), whereas the baseline models are distracted by superficially similar content (depicting generic ocean scenes). This failure stems from their embedding collapse, which blurs distinct events with similar global semantics. In contrast, by preserving fine-grained semantic structure, our approach disambiguates these contextually similar contexts and retrieves the exact segment corresponding to the query.

Query: "The camera is submerged in the water filming the ocean and divers."

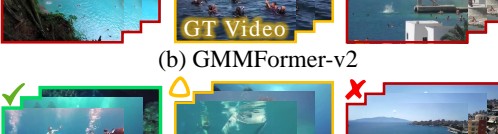

Figure 3: Retrieval example. 'GT Video' denotes the ground-truth paired video to the query. ✓, △, and ✗ indicate whether the retrieved video token is semantically aligned or not, regardless of its origin from the ground-truth video.

## 5    Conclusion & Limitation

**Conclusion.** In this paper, we address semantic collapse in PRVR, where semantically diverse text queries and video segments are undesirably attracted or repelled due to pairwise annotation schemes. To mitigate this, we propose a unified framework consisting of Text Correlation Preservation Learning (TCPL) and Cross-Branch Video Alignment (CBVA). TCPL distills the relational structure from CLIP to preserve semantic consistency across text queries, while CBVA aims to structure video embeddings according to their inherent semantics, supported by our token merging strategies. Extensive evaluations highlight the importance of addressing semantic collapse for effective PRVR.

**Limitation.** Our method has two limitations. First, as our method builds upon the pretrained CLIP model, it can inherit weaknesses; it may struggle with fine-grained spatial/directional queries (e.g., distinguishing "left of" from "right of"). However, we emphasize that this limitation does not extend to compositional understanding. As we demonstrate in the Appendix, our method actively corrects CLIP's common failure modes where the queries involve multi-entity contexts and multi-event temporal compositions (recovering 28% of CLIP's $R@1$ failure cases and 57% of its $R@10$ failure cases). Second, our framework incurs an increased training cost. However, for deployment, our model architecture does not introduce any new modules that increase inference time, incurring no additional latency compared to standard retrieval baselines.

## Acknowledgements

This work was supported in part by MSIT/IITP (No. RS-2022-II220680, RS-2020-II201821, RS-2019-II190421, RS-2024-00459618, RS-2024-00360227, RS-2024-00437633, RS-2024-00437102, RS-2025-25442569), MSIT/NRF (No. RS-2024-00357729), and KNPA/KIPoT (No. RS-2025-25393280).

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

Table A1: Sensitivity to temperature $\tau$ across datasets. Rows marked with gray indicate the default configuration used in the main results.

| Dataset | $\tau$ | R@1 | R@5 | R@10 | R@100 | SumR |
|---------|--------|-----|-----|------|-------|------|
| | 0.70 | 35.6 | 61.0 | 70.8 | 95.0 | 262.4 |
| | 0.75 | 35.5 | 61.2 | 71.1 | 94.9 | 262.6 |
| TVR | 0.80 | 35.1 | 61.6 | 71.5 | 94.9 | 263.1 |
| | 0.85 | 35.1 | 61.2 | 71.2 | 95.0 | 262.5 |
| | 0.90 | 35.1 | 61.1 | 71.1 | 94.9 | 262.2 |
| | 0.70 | 17.6 | 41.9 | 55.4 | 86.8 | 201.7 |
| | 0.75 | 17.8 | 41.9 | 55.4 | 86.7 | 201.8 |
| ANet | 0.80 | 17.7 | 42.0 | 55.6 | 86.8 | 202.1 |
| | 0.85 | 17.7 | 42.1 | 55.3 | 86.8 | 201.9 |
| | 0.90 | 17.2 | 41.9 | 55.5 | 86.8 | 201.4 |
| | 0.70 | 3.3 | 11.6 | 19.8 | 63.9 | 98.6 |
| | 0.75 | 3.4 | 12.7 | 19.4 | 64.8 | 100.3 |
| CHA | 0.80 | 3.4 | 12.0 | 18.7 | 64.5 | 98.6 |
| | 0.85 | 3.2 | 12.6 | 20.1 | 63.8 | 99.7 |
| | 0.90 | 3.3 | 12.4 | 19.1 | 64.0 | 98.9 |

## A    Further Analysis on Hyperparameter Sensitivity

We noted that all hyperparameters are unified across datasets except the similarity threshold $\tau$, which we set per dataset to account for different internal segment-to-segment similarity distributions [32]. Beyond the QVHighlights ablation, Table A1 evaluates $\tau$ sensitivity on TVR, ActivityNet-Captions (ANet), and Charades as well. Empirically, QVHighlights exhibits the lowest similarity levels, TVR and ANet are intermediate, and CHA shows the highest. Accordingly, we adopt $\tau=0.70$ for QVHighlights, $\tau=0.80$ for TVR and ANet, and $\tau=0.85$ for CHA. As shown, varying $\tau$ within a moderate range causes only minor fluctuations in each dataset, indicating that performance is not overly sensitive to this hyperparameter once set near the optimum.

## B    Impact of CLIP's Failure Rate on TCPL

In this section, we evaluate whether TCPL inherits or corrects CLIP's semantic errors in the PRVR setting. We conduct this study on the TVR dataset since most text queries in TVR involve multiple named entities or sequential actions that require the capability to comprehend complex temporal and contextual cues. On the test set of TVR (10,895 queries), we mark a success when the ground-truth video appears within the top-$Q$ retrieved results ($Q \in \{1, 10\}$) and compare our model (with TCPL) to zero-shot CLIP via a $2{\times}2$ outcome matrix. Specifically, for each text query, we record (i) both correct, (ii) ours correct & CLIP wrong, (iii) ours wrong & CLIP correct, and (iv) both wrong. Tab. A2 reports the counts (and proportions).

To illustrate, when $Q{=}1$, our model corrects 2,551 of CLIP's failures (while the reverse occurs in 500 cases); at $Q{=}10$, the corresponding counts are 3,627 vs. 386. Our proposed framework also retains CLIP's strengths, answering correctly together on 1,277 (R@1) and 4,162 (R@10) queries.

We further analyze the instances where one model succeeds and the other fails. When CLIP fails, the correct item is, on average, ranked 56th, indicating severe confusion. These failures consistently involve queries with multi-entity contexts and temporal compositions. For example, CLIP ranked the correct video at 237 for "Sebastian grabs his folder and stands up from the table" and at 418 for "George pulls back on Meredith's rolling chair and drags her". By contrast, when our model fails but CLIP succeeds, the ground-truth video is still ranked highly, with an average position of 6.7. These cases are typically simple and object-centric queries requiring little compositional or temporal reasoning. For instance, CLIP correctly retrieved the videos for "House takes a sip of soda from the bottle" and "Joey is folding his coat in the kitchen", while our model placed them at rank 2. Taken together, these outcomes demonstrate that the retrieval objective reshapes the representation toward task-specific temporal and compositional semantics, with TCPL preserving robust high-level alignment while correcting CLIP's fine-grained failure modes.

Table A2: Comparative analysis of retrieval correctness between our model and zero-shot CLIP on the TVR test set (10,895 queries), evaluated using (a) Recall@1 and (b) Recall@10 as success criteria. Values are raw counts with percentages in parentheses.

(a) Recall@1.

|  | CLIP correct | CLIP wrong |
|---|---|---|
| Ours correct | 1277 (11.7%) | 2551 (23.4%) |
| Ours wrong | 500 (4.6%) | 6567 (60.3%) |

(b) Recall@10.

|  | CLIP correct | CLIP wrong |
|---|---|---|
| Ours correct | 4162 (38.2%) | 3627 (33.3%) |
| Ours wrong | 386 (3.5%) | 2720 (24.9%) |

---

**Algorithm 1** Order-Preserving Token Merging (OP-ToMe)

---

**Require:** Frame tokens $V_f \in \mathbb{R}^{B_v \times L_f \times d_v}$, Merge rate $N\%$, Number of iterations $M$
**Ensure:** Clip tokens $V_c \in \mathbb{R}^{B_v \times L_c \times d_v}$ where $L_c = 32$
1:  Initialize token sizes $s \leftarrow \mathbf{1}_{L_f} \in \mathbb{R}^{L_f}$        ▷ Each token represents 1 frame
2:  **for** $m = 1$ to $M$ **do**
3:      Compute cosine similarity between disjoint adjacent-frame pairs:
         $S[i] \leftarrow \cos(V_f[i], V_f[i+1])$ for $i = 1, 3, 5, \ldots, L_f - 1$
4:      Select top-$N\%$ most similar adjacent pairs based on $S$
5:      **for** each selected pair $(i, i+1)$ **do**
6:          Compute size-weighted average:
         $V_{\text{merged}} \leftarrow \frac{s[i] \cdot V_f[i] + s[i+1] \cdot V_f[i+1]}{s[i] + s[i+1]}$
7:          Replace $V_f[i]$ with $V_{\text{merged}}$, remove $V_f[i+1]$
8:          Update size: $s[i] \leftarrow s[i] + s[i+1]$, remove $s[i+1]$
9:      **end for**
10:     Update $L_f \leftarrow$ new token length
11:     **if** $L_f \leq 32$ **then**
12:         **break**
13:     **end if**
14: **end for**
15: **return** $V_c \leftarrow V_f$

---

## C    Algorithms for Cross-Branch Video Alignment

In this section, we provide a detailed algorithm for sub-components of our Cross-Branch Video Alignment (CBVA). Particularly, we illustrate Order-Preserving Token Merging (OP-ToMe), the process of pre-computing a discrete set of different levels of clip number (number of semantics), and the process of per-video merging for Adaptive CBVA in Algorithm. 1, Algorithm. 2, and Algorithm. 3, respectively.

## D    Positive and Negative Societal Impacts

**Positive Impact.** Our work improves the text-video retrieval based on partial content descriptions within long, untrimmed videos. We expect that the proposed method will enhance the user experience in video search and navigation. This is particularly valuable in domains such as education, where lengthy untrimmed videos are commonly utilized.

**Negative Impact.** However, the ability to isolate specific video contexts and retrieve segments based on partial descriptions could be misused in surveillance settings (e.g., CCTV), enabling the tracking of individuals or the extraction of sensitive behaviors without consent. Such misuse may raise potential concerns regarding privacy and ethical deployment.

**Algorithm 2** Pre-computing the different levels of clip number (Eq. 9)

---

**Require:** Initial clip length $L_c^1 = L_c$ (e.g., 32), merge-rate $N\%$, minimum clips $C_{\min}$
**Ensure:** Candidate list $L = \left[ L_c^1, L_c^2, \ldots, L_c^K \right]$
 1: $i \leftarrow 1, \quad L \leftarrow \left[ L_c^1 \right]$
 2: **while** $L_c^i > C_{\min}$ **do**
 3: $\quad L_c^{i+1} \leftarrow \max\!\left( 2 \times \left\lfloor \dfrac{L_c^i - (L_c^i/2)\,(N/100) + 1}{2} \right\rfloor, \; C_{\min} \right)$
 4: $\quad$ **if** $L_c^{i+1} = L_c^i$ **then break**
 5: $\quad$ **end if**
 6: $\quad$ Append $L_c^{i+1}$ to $L$
 7: $\quad i \leftarrow i + 1$
 8: **end while**
 9: $K \leftarrow |L|$ $\qquad\qquad\qquad\qquad\qquad\qquad\qquad\qquad$ ▷ number of discrete clip levels
10: **return** $L$

---

**Algorithm 3** Constructing merged clips for Adaptive CBVA

---

**Require:** Clip tokens $V_c \in \mathbb{R}^{B_v \times L_c \times d_v}$, Global candidate list $L$ of length $K$, Merge rate $N\%$, Similarity threshold $\tau$, Projected Clip tokens $\bar{V}_c \in \mathbb{R}^{B_v \times L_c \times d}$,
**Ensure:** Adapted clip tokens $\tilde{V}_c$ with length $L_c^*$
$\quad$ **Stage 1. Estimate internal similarity**
 1: Compute cosine-similarity matrix $S$ from *frozen* $V_c$
 2: $\omega \leftarrow \dfrac{\left| \{(i,j) : S_{ij} > \tau, \; i \neq j\} \right|}{L_c(L_c - 1)}$ $\qquad\qquad\qquad\qquad$ ▷ high-similarity ratio
$\quad$ **Stage 2. Select merging depth** $k^*$
 3: **if** $\omega \leq 1 - \frac{1}{K}$ **then** $\qquad\qquad\qquad\qquad\qquad\qquad$ ▷ if diverse, keep all clips
 4: $\quad k^* \leftarrow 1$
 5: **else**
 6: $\quad k^* \leftarrow \min_{k \in \{2, \cdots, K\}} (w > \frac{K-k}{K})$
 7: **end if**
$\quad$ **Stage 3. Merge clips** $k^* - 1$ **times**
 8: $\tilde{V}_c \leftarrow \bar{V}_c$
 9: **for** $m = 1$ **to** $k^* - 1$ **do**
10: $\quad$ Apply *bipartite token merging (TOME)* [1] to $\tilde{V}_c$ at rate $N\%$
11: **end for**
12: $L_c^* \leftarrow |\tilde{V}_c|$
13: **return** $\tilde{V}_c$

---

