# OpenReview forum: "Mitigating Semantic Collapse in Partially Relevant Video Retrieval"
_NeurIPS.cc/2025/Conference — NeurIPS 2025 poster_

### Official Review · Reviewer_3bLp · 2025-06-30

**Clarity:** 2
**Significance:** 3
**Originality:** 3
**Rating:** 4
**Confidence:** 3

**Summary:**

In this paper, authors propose a novel method for PRVR, which addresses the semantic collapse problem on the text and video side.
The proposed method contains two modules, *i.e.*, CBVA and TCPL. Extensive experiments show the superiority of the proposed method.

**Questions:**

See weakness.

**Ethical Concerns:**

["NO or VERY MINOR ethics concerns only"]

**Final Justification:**

Authors' response addressed most of my concerns. However, this paper to me is an okay/good enough paper. So I maintain my rating at 4 (Leaning to accept but not a clear accept, borderline accept).

**Limitations:**

yes.

**Paper Formatting Concerns:**

No.

**Quality:**

3

**Strengths And Weaknesses:**

Strengths:
1. The proposed method is novel to me.
2. The proposed model consistently outperforms state-of-the-art methods on four PRVR datasets with up to 8-point gains in SumR.
3. Demonstrates the necessity of each component (TCPL, OP-ToMe, adaptive CBVA) with clear metrics and diagnostic experiments.

Weakness:
1. Line 126 - Line 133. Confirm that the frame-level loss only use one visual token per video for l_f^* right?
2. What is the \delta for huber loss (Line 163)?
3. How do \lambda^E and \lambda^A affect performance?
4. Line 192. How to get the size-weighted average?
5. What backbones do methods in Tables 4 and 5 use?
6. While L^base explore frame-level loss, how about work-level loss?
7. What if the benchmark dataset does only have one-to-one video-text data pair?
8. For now, the proposed methods try to address the problem on the uni-modal side, text and video sides. How about cross-modal side? For example, frame-to-word?

---

> ### Author Rebuttal · Authors · 2025-07-30
>
> # Only one visual token per video is used to calculate $\mathcal{L}_{f}^{*}$?
> Yes, the reviewer is correct.
> To compute $\mathcal{L}_{f}^{*}$, we select only the video token (from frame-level encodings) that yields the highest similarity to the given text query.
> This design choice reflects the nature of long untrimmed videos, where only a specific temporal region typically aligns with the query.
> By focusing on the most relevant token, we encourage the model to learn precise and semantically meaningful associations, rather than diffusing supervision across irrelevant video segments.
>
> # What is delta for Huber loss?
> We apologize for the confusion.
> In our custom implementation, we do not use the Huber‑loss parameter $\delta$; it is fixed at the standard default value of 1.
>
> # How do $\lambda^{E}$ and $\lambda^{A}$ affect performance?
> As shown in Tab. 3 (a,b), the TCPL loss consistently improves performance across a range of values.
> This demonstrates that our method is robust to the specific values of $\lambda^{E}$ and $\lambda^{A}$ as long as they are within a reasonable scale.
> We selected these weights using general heuristics without tuning them per dataset to avoid overfitting.
> We will clarify this robustness in the revised manuscript.
>
> # L192. How to get the size-weighted average?
> We follow the original Token Merging (ToMe) paper [A] and adopt a size-weighted averaging procedure during token merging.
> Specifically, each visual token is assigned a size vector that is initially set to 1, representing a single frame.
> When two tokens $\mathbf{a}$ and $\mathbf{b}$ are merged, their corresponding size values $n_a$ and $n_b$ are summed to reflect how many original tokens they now represent.
> The merged token $\mathbf{m}$ is then computed as the weighted average of the two token embeddings:
> \begin{equation}
>     \mathbf{m} = \frac{n_a \times \mathbf{a} + n_b \times \mathbf{b}}{n_a + n_b},
> \end{equation}
> and its size is updated to the sum of their sizes, i.e., $n_m = n_a + n_b$.
> This operation ensures that the resulting token faithfully aggregates semantic information from the constituent frames in proportion to their temporal extent.
>
> [A] Bolya et al. Token Merging: Your ViT But Faster. ICLR 2023.
>
> # What backbones do methods in Tab. 4 and Tab. 5 use?
>
> All methods compared in Tab. 4 and Tab. 5 use identical backbones to ensure fair comparisons across all baselines.
> The list of backbones used for all methods is provided below.
>
> | Dataset | Visual | Text |
> |--|--|--|
> | QVHighlights | CLIP-B/16 + Slowfast | CLIP-B/16 |
> | TVR | CLIP-L/14 | CLIP-L/14 |
> | ANet-Captions | CLIP-L/14 | CLIP-L/14 |
> | Charades-STA | CLIP-L/14 | CLIP-L/14 |
>
> # While $L^\text{base}$ explores frame-level loss, how about work-level loss?
>
> We interpret this as a suggestion to explore word-level (i.e., token-wise) loss for finer-grained text supervision.
>
> However, there are two primary reasons we opted for sentence-level loss:
>
> 1) Computational Efficiency: Word-level alignment significantly increases training cost since our Text Correlation Preservation Learning (TCPL) involves both pairwise similarity calculation for Euclidean distance and triplet similarity calculation for angular distance.
>
> 2) Robustness to Domain Variance: Sentence-level embeddings are more stable across domains, whereas word-level tokens can vary significantly depending on the context.
> For instance, in real-world video datasets like TVR used in Partially Relevant Video Retrieval (PRVR), text often includes domain-specific proper nouns (e.g., person names or locations).
> Such variability can make word-level alignment less reliable and prone to overfitting. By focusing on [EOS] token supervision, we encourage the model to capture more domain-invariant semantics.
>
> # What if the benchmark dataset does only have one-to-one video-text data pair?
>
> This situation can be interpreted in two ways:
>
> 1) When each video is still long and untrimmed, so that videos contain multiple semantic contexts internally, but only one query is provided per video:
>
> In this case, although assigning only one query per video removes intra-video semantic confusion to some extent, semantic collapse can still occur due to semantic overlaps across videos.
> To verify the effectiveness in this scenario, we trained our baseline and proposed method using one query per video and evaluated it on the full QVHighlights test set.
> Results are as below:
>
> | QVHighlights | R1 | R5 | R10 | R100 | SumR |
> |--|--|--|--|--|--|
> | Base | 13.1 | 32.1 | 44.7 | 86.7 | 176.6 |
> | Ours | 15.6 | 34.6 | 47.5 | 90.1 | 187.8 |
>
> As shown, even in the one-to-one setting, our method demonstrates clear improvements, highlighting its robustness and general applicability to PRVR-style scenarios.
> The overall relative performance drop compared to Tab. 4 in the main paper is primarily due to the drastic reduction in training pairs (only about 30\% of the original text-video pairs are retained under the one-to-one pairing constraint).
>
> 2) When each video corresponds to a single semantic concept and is fully matched by one query:
>
> In this case, the setup reduces to standard video-text retrieval where videos are short and semantically coherent (i.e., aligned one-to-one with a query).
> Thus, the risk of semantic collapse is significantly reduced.
> Although our method would still function properly in such settings, we expect less contribution from CBVA or TCPL since our method is specifically designed for PRVR scenarios, where videos are long, untrimmed, and associated with multiple semantics.
>
> To illustrate, Cross‑Branch Video Alignment (CBVA) assumes that a single video may contain several distinct semantic segments (PRVR scenario) and learns to explicitly distinguish different semantics within the long untrimmed videos.
> On the other hand, TCPL tackles the PRVR-specific problem where multiple queries tied to the same video are forced together (while semantically similar queries from other videos are pushed apart).
> Therefore, our gains are expected to be naturally smaller in the one-to-one setting, where videos are matched fully with the paired query.
> Again, this is not because the method fails, but because the risk of semantic collapse is significantly reduced.
>
>
> # For now, the proposed methods try to address the problem on the uni-modal side, text and video sides. How about cross-modal side? For example, frame-to-word?
>
> As the reviewer mentioned, our work primarily focuses on addressing semantic collapse on uni-modal side.
> Particularly, we leveraged underexplored yet reliable forms of supervision (the semantic understanding capability of the vision-language backbone and video timestamps to construct self-supervised objectives that capture intra-video semantic differences).
>
> We thank the reviewer for the insightful suggestion.
> We agree that addressing semantic collapse from a cross-modal perspective is a promising future direction.
> In fact, we experimented with applying TCPL to cross-modal pairs (e.g., sentence-to-frame and word-to-frame alignment), but it led to suboptimal results.
> We believe this is primarily due to domain shifts between the visual content and the text vocabulary, which makes CLIP's text-video aligned space suboptimal for PRVR (as can be inferred from CLIP's low zero-shot performance in Tab. 5).
> Therefore, designing an effective cross-modal strategy requires more careful formulation than merely distilling CLIP's knowledge and remains an important avenue for future work.
> As future work, we intend to develop more robust cross-modal alignment techniques that can better adapt to these domain discrepancies.

---

> > ### Comment · Reviewer_3bLp · 2025-08-05
> >
> > Thank the authors for the response. It addressed most of my concerns. I have updated correspondingly.

---

> ### Author Response · Authors · 2025-08-06
>
> Dear Reviewer 3bLp
>
> We sincerely appreciate the time you took to read our rebuttal and for updating your review.
> We are glad that our additional clarifications resolved your concerns.
>
> If any further questions arise during the discussion period, please let us know.
> We will be happy to provide more details.
>
> Thank you again for your constructive feedback and for helping us strengthen the paper.

---

### Official Review · Reviewer_jkfJ · 2025-07-02

**Clarity:** 1
**Significance:** 3
**Originality:** 3
**Rating:** 4
**Confidence:** 4

**Summary:**

This paper focuses an interesting problem in partially relevant video retrieval, where existing loss functions may inadvertently push apart semantically similar features while pulling together dissimilar ones, i.e., "semantic collapse" called by the authors. To mitigate this, the authors propose two constraints: TCPL (Section 3.2) and CBVA (Section 3.3) to preserve the learned feature representation consistent with the original CLIP semantic space.
Although this is an interesting question, the proposed method does not solve this problem well. The performance improvement may be achieved at the expense of the model's computational efficiency and inference speed.

**Questions:**

1. The paper needs to discuss computational efficiency or inference latency, especially compared to exiting works.  The merging operation in Section 3.3 appears computationally slow, yet no runtime analysis is provided.
2.  Training objective in Section 3.3, that pushes apart features from different timestamps of the same video, seems unreasonable.
 Cause videos typically maintain semantic consistency across timestamps.

**Ethical Concerns:**

["NO or VERY MINOR ethics concerns only"]

**Final Justification:**

Thanks for your response, and most of my concerns are addressed. I tend to raise my score.

**Limitations:**

Authors should make the writing and figures clearer. Figure 2 makes the simple distillation too complicated and difficult to read. Similarly, Figure 1 is also too complicated. More critically, as the core chapter of this article, the process of adaptive CBVA in section 3.3 is too complicated, making it difficult to understand how the method is actually implemented.

**Paper Formatting Concerns:**

Not applied

**Quality:**

2

**Strengths And Weaknesses:**

1. The paper makes a valuable observation about current partially relevant video retrieval methods: existing loss functions may inadvertently push apart semantically similar features while pulling together dissimilar ones (semantic collapse). This problem formulation is clearly presented.
2. Section 3.1 is well-written. However, the key Section 3.3 lacks sufficient methodological clarity.

---

> ### Author Rebuttal · Authors · 2025-07-29
>
> # Computational efficiency \& Inference latency
> To first clarify efficiency concerns, we first note that our baseline is built upon GMMFormer V2, but excludes the two objectives proposed in that work: (1) revamped query diversity loss and (2) optimal matching loss.
> Thus, our baseline maintains the same model scale as GMMFormer V2.
> In addition, we report inference time, inference memory, training time, training memory, the number of parameters, and FLOPs on the QVHighlights benchmark.
> Inference time and training time are averaged over five independent runs.
>
> (1) Inference time & memory : Across database sizes from 100 to 474 videos in QVHighlights, ours offers the second‑best inference latency and memory footprint, while achieving substantially higher retrieval accuracy.
> Note that video features are pre-computed and stored offline in real-world text-video retrieval systems; when a user inputs a text query, the system operates on these cached features. As a result, our token-merging step imposes no additional cost at inference time.
>
> (2) Training time & memory : Although our training incurs greater time and memory to learn fine-grained video contexts, we argue that the training cost is paid offline.
> Instead, real-world deployment is driven by efficiency during inference.
>
>
> |Inference time(ms)|100 (vid DB size)|200|300|400|474|
> |--|--|--|--|--|--|
> |MSSL|3.09|3.85|4.66|5.14|5.58|
> |GMM|1.97|1.98|1.99|2.02|2.05|
> |GMMv2|2.31|2.38|2.40|2.61|2.78|
> |Ours|2.32|2.37|2.40|2.60|2.70|
>
> |Inference Mem(MB)|100|200|300|400|474|
> |--|--|--|--|--|--|
> |MSSL|717.47|796.15|874.83|954.14|1010.89|
> |GMM|243.11|248.95|254.78|260.62|264.10|
> |GMMv2|419.75|440.18|459.62|480.55|493.46|
> |ours|419.75|440.18|459.62|480.55|493.46|
>
> |Others|MSSL|GMM|GMMv2|Ours|
> |--|--|--|--|--|
> |Train Time 1 epoch(ms)|10934|12828|17223|62641|
> |Train Mem(MB)|2375|3333|7826|9755|
> |Model Param(m)|4.57|12.72|32.14|32.14|
> |Inference FLOPs(G)|0.37|0.99|2.78|2.78|
>
> # Justification for Cross-Branch Video Alignment
> We completely agree that blindly pushing apart features from different timestamps within the same video is unreasonable.
> In fact, as shown in Tab. 1(c), the naive CBVA (which repels all cross-timestamp pairs), yields little to no improvement even on PRVR benchmarks that involve multiple semantics per video.
>
> To address this limitation, our adaptive CBVA explicitly avoids such indiscriminate repulsion.
> Before applying the CBVA loss, we first perform a token merging step that semantically groups relevant segments, even if they are temporally distant (L212-214).
> Consequently, our final adaptive CBVA only pushes apart frame–clip pairs that (1) are from distinct timestamps and (2) exhibit low semantic similarity.
> This ensures the model learns to discriminate between different events while preserving semantically coherent ones.
>
> This ensures that our final CBVA promotes meaningful separation between distinct events, while preserving and aligning semantically coherent segments regardless of their timestamp.
>
> Furthermore, to enhance the readability of Adaptive CBVA, we will revise L214-218 as "However, optimizing the number of semantics per video is costly during the token merging process.
> Therefore, we instead pre-define a discrete set of clip numbers based on a fixed merge rate, and then match each video to the level that best reflects its internal similarity structure (number of different semantics).
> To initially establish a discrete set of clip levels, we define $N\%$ to denote the merge rate and $C_\text{min}$ to represent the minimum number of semantically different clips in each video.
> Then, we generate $K$ levels of clip number candidates {${L_c^i}$}$_{i=1}^{K}$ by recording clip number after each merge step as:".
>
> # Methodological / Figure Clarity
> We appreciate the reviewer’s positive feedback on Section 3.1 and acknowledge the need for clearer exposition in Section 3.3.
> To address this, we provide step-by-step algorithms below describing the OP-ToMe and adaptive CBVA process, and simplify the paragraph structure for better readability.
> These will be added to Appendix.
> ────────────────────────────────────────────────────────────────────────────
>
> ### Algorithm 1. Order‑Preserving Token Merging (OP‑ToMe)
> ```
> Input : Frame tokens  V_f ∈ ℝ^{B_v × L_f × d_v}
>         Merge rate    N %      (percentage of pairs to merge per iteration)
>         Iterations    M
> Output: Clip tokens   V_c ∈ ℝ^{B_v × L_c × d_v}   where L_c = 32
> ```
> ```
> 1.  Initialize token‑size vector  s ← 1_{L_f}        # each token covers 1 frame
>
> 2.  for m = 1 … M do
> 3.      # 1. Compute similarity of disjoint adjacent pairs (odd indices 1,3,5,…)
> 4.      for i ∈ {1, 3, 5, … , L_f‒1}:
> 5.          S[i] ← cos( V_f[i], V_f[i+1] )
> 6.      end for
> 7.
> 8.      # 2. Pick the top‑N% most‑similar adjacent pairs
> 9.      P ← indices of top‑N pairs in S
> 10.
> 11.     # 3. Merge each selected pair
> 12.     for (i, i+1) ∈ P (processed left→right):
> 13.         V_merge ← ( s[i]·V_f[i]  +  s[i+1]·V_f[i+1] ) / ( s[i] + s[i+1] )
> 14.         V_f[i]← V_merge                 # replace left token by the merge
> 15.         delete V_f[i+1]                   # remove right token
> 16.         s[i] ← s[i] + s[i+1]            # update size
> 17.         delete s[i+1]
> 18.     end for
> 19.
> 20.     L_f ← |V_f|                           # new token length
> 21.     if L_f ≤ 32 then break
> 22. end for
> 23.
> 24. return V_c ← V_f                          # 32 order‑preserving clip tokens
> ```
> ────────────────────────────────────────────────────────────────────────────
>
> ### Algorithm 2. Pre-computing the discrete clip-count levels before Adaptive CBVA
> ```
> Input : Initial clip length   L_c^1 = L_c      (e.g., 32)
>         Merge‑rate            N %              (percentage applied per step)
>         Minimum clip count    C_min
> Output: Candidate list        L = [L_c^1 , L_c^2 , … , L_c^K]
> ```
> ```
> 1.  i ← 1                                   ;  L ← [ L_c^1 ]
> 2.  while  L_c^i > C_min  do
> 3.      # Compute next coarser clip length
> 4.      L_new ← max( 2 × floor( ( L_c^i − (L_c^i /2) × (N/100) + 1 ) / 2 ),
> 5.                     C_min )
> 6.      if  L_new = L_c^i   then  break      # no further reduction possible
> 7.      append  L_new  to  L
> 8.      i ← i + 1
> 9.  end while
> 10. K ← |L|                                  # number of discrete levels
> 11. return  L
> ```
> ────────────────────────────────────────────────────────────────────────────
>
> ### Algorithm 3. Constructing Merged Clips for Adaptive CBVA
> ```
> Input : Clip tokens        V_c  ∈ ℝ^{B_v × L_c × d_v}
>         Global candidate   L    = [L_c^1 … L_c^K]   # pre‑computed clip counts
>         Merge‑rate         N%                      # applied at each merge
>         Similarity thresh. τ
> Output: Adapted clip tokens  Ṽ_c   (length  L_c* )
> ```
> ```
> # Stage 1 – Estimate internal similarity
> 1.  S ← cosine‑similarity matrix of frozen V_c              # L_c × L_c
> 2.  ω ← |{(i,j) : S_ij > τ , i≠j}| / [L_c (L_c − 1)]        # high‑sim ratio
>
> # Stage 2 – Select merging depth k*
> 3.  if  ω ≤ 1 − 1/K                     then   k* ← 1        # keep all clips
> 4.  else
> 5.       k* ← min{ k ∈ {2 … K} |  ω > (K − k)/K }           # pick coarser level
> 6.  end if
>
> # Stage 3 – Merge clips (k* − 1 times)
> 7.  Ṽ_c ← V_c
> 8.  for  m = 1 … (k* − 1)  do
> 9.        Apply bipartite token‑merging (ToMe) to Ṽ_c at rate N %
> 10. end for
> 11. L_c* ← |Ṽ_c|
> 12. return  Ṽ_c
> ```
>
> We also appreciate the feedback on visual clarity.
> For Fig.1, we agree that the figure may seem too complicated and may obscure the core message.
> To improve clarity, we will revise the color coding of the circles with our core message; instead of depicting generic positive/negative relations, we will use green to indicate text–video pairs that share similar semantics but belong to different videos (thus being pushed apart), and red to indicate segments with different semantics within the same video (thus mistakenly pulled together).
> This change will better illustrate the misalignment issues addressed by our method.
> For Fig.2, we will simplify the illustration by removing the explicit annotations of relation types (e.g., positive/negative edges) and focus solely on conveying the core idea of relation distillation.
> We will also add short description such as “Distill the relation” to emphasize the key message.
> Additionally, we will include a new illustrative diagram specifically for adaptive CBVA and annotate each merging step.
> Overall, these changes will significantly enhance readability and understanding.

---

> > ### Author Response · Authors · 2025-08-06
> >
> > Dear Reviewer jkfJ
> >
> > Thank you very much for your detailed and constructive comments.
> >
> > We have prepared a point-by-point response that clarifies **(1)** Computation efficiency, **(2)** Justification for Cross-Branch Video Alignment, **(3)** Clarification for our method, and a plan to clarify Fig. 1.
> >
> > We would be grateful to hear whether these additions address your concerns, or if any particular aspect would benefit from further elaboration or experiments.
> >
> > We are happy to supply more details during the discussion phase.
> >
> > Thank you again for your time and helpful feedback.
> >
> > Authors

---

> ### Comment · Reviewer_jkfJ · 2025-08-07
>
> Thanks for your response, and most of my concerns are addressed. I tend to raise my score.

---

> ### Author Response · Authors · 2025-08-08
>
> Dear Reviewer jkfJ,
>
> We sincerely thank you for your time and thoughtful engagement in reviewing our rebuttal and for updating your assessment accordingly. We are pleased to know that our clarifications have sufficiently addressed your concerns.
>
> Should any further questions or points of discussion arise during the discussion period, please do not hesitate to let us know. We would be glad to provide additional clarification as needed.
>
> Once again, we are grateful for your constructive feedback and for your contribution to strengthening the quality of our paper.

---

### Official Review · Reviewer_aktj · 2025-07-06

**Clarity:** 2
**Significance:** 2
**Originality:** 2
**Rating:** 3
**Confidence:** 5

**Summary:**

This paper addresses semantic collapse in Partially Relevant Video Retrieval (PRVR), categorized into: (i) text semantic collapse, where queries from the same video cluster despite semantic irrelevance, while similar queries across videos are separated; and (ii) video semantic collapse, where segments from the same video converge regardless of semantic disparity. To mitigate (i), the authors propose TCPL to distill semantic relations from CLIP. For (ii), they introduce CBVA to align embeddings of temporally matched frames and clips while separating others. Additionally, OP-ToMe is proposed for various clip embeddings. Extensive experiments on four datasets validate the model’s superior retrieval performance.

**Questions:**

1. Why are the ablation studies conducted only on QVHighlights? How about the ablation results on other datasets?

2. Could the authors elaborate on possible failure scenarios, particularly in the context of highly complex videos—such as those involving frequent context shifts, significant visual ambiguity, or high levels of noise—where token merging or adaptive CBVA may struggle to produce accurate segmentation? Additionally, are there any mitigation strategies that could be employed in such cases?

3. TCPL heavily relies on the capabilities of the CLIP model, which is known to exhibit biases in fine-grained semantics (e.g., “picking up a puppy” vs. “petting a puppy”) and rare concepts. If CLIP itself incorrectly models the relationship between certain query pairs—such as misclassifying semantically related pairs as unrelated—would TCPL amplify such errors? Could the authors provide ablation studies to assess the impact of CLIP’s error rate on TCPL’s performance?

4. Could the authors explain in more detail what the "baseline" model is used in this paper? I noticed that the results reported in Table 1 and Table 4 show that the baseline already achieves strong performance, with marginal differences (1.1 SumR, ~0.3 average) behind ProtoPRVR. Are the performance gains reported in the paper primarily attributed to the strong baseline itself?

5. In Equations 4 and 5, why is the [EOS] token chosen for distillation? Additionally, at which stage is OP-ToMe applied—before feeding inputs into the clip-level branch, or at the output layer?

**Ethical Concerns:**

["NO or VERY MINOR ethics concerns only"]

**Final Justification:**

While minor points remain, the authors' revisions have sufficiently improved the manuscript to a borderline level.

**Limitations:**

Yes.

**Quality:**

3

**Strengths And Weaknesses:**

**Strengths**
1. The paper is well-formatted, and Figure 1’s design and color scheme are visually appealing and conceptually clear.
2. The definitions of semantic collapse and the corresponding proposed method are clear and easy to understand.
3. The method performs well on PRVR retrieval benchmarks.


**Weaknesses**

1. The technical contributions are somewhat incremental and combinational. The core components, such as CLIP-based structural distillation and token merging, themselves are adaptations or applications of existing techniques rather than novelties.

2. The proposed method introduces more than 7 additional hyperparameters to tune, raising concerns about the method’s generalizability and applicability in deployment.  It remains unclear whether the reported performance gains are due to overfitting to the test set. The paper did not mention whether the experiments have independent validation and test sets.

3. Efficiency analysis is missing. The proposed method leverages a heavier backbone and employs multi-iteration aggregation to generate clip embeddings, which raises efficiency concerns. Therefore, it is necessary to report model size and provide any efficiency analysis, such as training-time cost, inference-time retrieval speed, or memory overhead, to justify the trade-off.

4. Missing comparison with baselines [A-D].

----

[A] Yin, Shukang, et al. "Exploiting Instance-level Relationships in Weakly Supervised Text-to-Video Retrieval." ACM Transactions on Multimedia Computing, Communications and Applications 20.10 (2024): 1-21.

[B] Zhang, Qun, et al. "Multi-Grained Alignment with Knowledge Distillation for Partially Relevant Video Retrieval." ACM Transactions on Multimedia Computing, Communications and Applications (2025).

[C] Jiang, Xun, et al. "Progressive Event Alignment Network for Partial Relevant Video Retrieval." 2023 IEEE International Conference on Multimedia and Expo (ICME). IEEE, 2023.

[D] Song, Peipei, et al. "Towards Efficient Partially Relevant Video Retrieval with Active Moment Discovering." arXiv preprint arXiv:2504.10920 (2025). Accepted by TMM 2025.

---

> ### Author Rebuttal · Authors · 2025-07-29
>
> # Technical Contributions incremental.
> We respectfully disagree with the characterization that our contributions are “merely incremental”.
> Our work introduces a comprehensive formulation of semantic collapse in PRVR (spanning both text & video modalities), quantifies it (Tab 6,7), and mitigates it with purpose-built objectives.
> These elements go beyond simply combining existing tools and address facets that prior art has either overlooked or only partially addressed.
>
> To address text semantic collapse incurred by determining relationships between text queries with matched video IDs:
>
> - Proposed TCPL, which distills relational structure from CLIP to preserve semantic distinctiveness across related queries.
> Unlike prior methods that simply push every same‑video query apart, TCPL achieves strong performance gain by preserving meaningful structure of the shared embedding space.
>
> To address video semantic collapse caused by overlapping events in untrimmed videos:
>
> - Proposed novel CBVA, which aligns hierarchical frame and clip-level features to resolve intra-video ambiguity.
> OP‑ToMe is tailored for temporal coherence in untrimmed videos: unlike generic token‑merging schemes, it merges only consecutive frames, thereby respecting playback order.
> Adaptive CBVA dynamically estimates video context granularity to better align hierarchical representations based on video-specific semantic density.
>
> # Hyperparams & Overfitting.
> We acknowledge the use of several hyperparams (e.g., TCPL weights ($\lambda^{\text{E}}$, $\lambda^{\text{A}}$), CBVA weight ($\lambda^{\text{CBVA}}$), merge rate ($N\%$), similarity threshold ($\tau$), and the minimum clip count ($C_{\text{min}}$)).
> Yet, we emphasize that our hyperparams(except $\tau$) were held constant across all datasets to avoid overfitting and ensure fair comparisons.
> To clarify our hyperparam selection strategy, we provide the following justifications based on empirical observations:
>
> $\lambda^{\text{E}}$, $\lambda^{\text{A}}$: As in Tab.3(a,b) in paper, TCPL consistently outperforms baseline (225.5 SumR) across a range of settings, as long as the values are within a reasonable scale.
>
> $\lambda^{\text{CBVA}}$, $N\%$: In Tab.3(d,e), we report consistent performance gains with intuitive values.
> Also, the same hyperparams are used for other datasets, demonstrating that the CBVA hyperparams generalize well and are not overfitted to any specific dataset.
>
> $\tau$: As shown in Tab.3(f), varying $\tau$ in the range of 0.5 to 0.8 yields stable and effective results, again indicating generalizability across datasets.
>
> $C_{\text{min}}$: This was heuristically set to 5 across all datasets, reflecting the typical upper bound of text queries per video observed in PRVR benchmarks.
>
> - These findings suggest that ours **does not rely on hyperparam tuning and avoids overfitting to specific datasets**.
> In contrast, we note that **our baselines** GMM&GMMv2 use **dataset-specific hyperparams while our method consistently outperforms them**.
>
> # Baseline & Efficiency & Complexity
> We apologize for any confusion caused regarding our baseline.
> To clarify, our baseline is derived from GMM v2, excluding its two key contributions: (1) revamped query diversity loss and (2) optimal matching loss; in terms of model size, our backbone remains on the same scale as GMM v2, and we adopt its hyperparams without additional tuning.
> Thus, we claim that our gains are not from the strong baseline but our proposed components that yield 9.1 SumR gain on QVHighlights, while objectives in GMMv2 are not as effective with CLIP backbone.
>
> Also, for efficiency, we provide time and memory for both inference and training along with # of params, and FLOPs on QVH.
> Inference and training time are averaged over 5 runs.
>
> (1) Inference time&mem: Across DB sizes from 100 to 474 videos, our method delivers the second‑best inference latency and memory footprint, while achieving substantially higher retrieval accuracy.
> Note that video features are pre-computed and stored offline in real-world text-video retrieval systems; when a user inputs a text query, the system operates on these cached features.
> As a result, our token-merging step imposes no additional cost at inference time.
>
> (2) Train time&mem: We acknowledge our training incurs more time and memory costs to learn fine-grained video contexts.
> Yet, we argue that training is performed offline; inference efficiency drives real-world deployment.
>
> |Inference time(ms)|100 (vid DB size)|200|300|400|474|
> |--|--|--|--|--|--|
> |MSSL|3.09|3.85|4.66|5.14|5.58|
> |GMM|1.97|1.98|1.99|2.02|2.05|
> |GMMv2|2.31|2.38|2.40|2.61|2.78|
> |Ours|2.32|2.37|2.40|2.60|2.70|
>
> |Inference Mem(MB)|100|200|300|400|474|
> |--|--|--|--|--|--|
> |MSSL|717.47|796.15|874.83|954.14|1010.89|
> |GMM|243.11|248.95|254.78|260.62|264.10|
> |GMMv2|419.75|440.18|459.62|480.55|493.46|
> |Ours|419.75|440.18|459.62|480.55|493.46|
>
> |Others|MSSL|GMM|GMMv2|Ours|
> |--|--|--|--|--|
> |Train Time 1 epoch(ms)|10934|12828|17223|62641|
> |Train Mem(MB)|2375|3333|7826|9755|
> |Param(m)|4.57|12.72|32.14|32.14|
> |FLOPs(G)|0.37|0.99|2.78|2.78|
>
> # Missing baseline
> We thank the reviewer for noting the missing baselines (will be referenced).
> Below shows reproduced scores with official codes when available; our method outperforms these by large margins(will be added to paper).
>
> | |QVH|ANET|CHA|TVR|
> |--|--|--|--|--|
> |BGM-Net|20.6 46.3 58.8 94.0 219.7|15.6 37.9 51.3 85.4 190.3|3.0 11.8 18.2 63.7 96.7|31.1 56.3 66.5 93.8 247.7|
> |MGAKD|X|X|X|X|
> |PEAN|X|X|X|X|
> |AMDNet|17.4 40.8  55.0  93.4  206.6|14.0 36.3 49.9 84.2 184.5|2.1 7.8 13.9 57.2 81.1|27.7 52.3 63.3 92.3 235.6|
> |Ours|23.9 51.5 63.7 95.5 234.6|17.7 42.0 55.6 86.8 202.1|3.2 12.6 20.1 63.8 99.7|35.1 61.6 71.5 94.9 263.1|
>
> # Why ablation on QVH?
> We concentrated our ablations on QVH since (1) its untrimmed videos exhibit frequent and diverse context changes, making it the most demanding benchmark, and (2) it is built from a recent crawl of YouTube content, so its distribution closely mirrors real-world video-retrieval scenarios.
>
> To confirm that the gains are not dataset‑specific, we conducted the ablation on ANET.
> Improvement pattern is almost identical, showing that our components generalize well across datasets.
> Notably, since videos in ANET contain fewer intra‑video scene changes than QVH, the text‑side (TCPL) contribution is bigger than video‑side gains (as can be expected when visual context diversity is lower).
>
> |Method|R1|R5|R10|R100|SumR|
> |--|--|--|--|--|--|
> |Base|16.1|39.5|52.7|85.9|194.2|
> |+TCPL|17.2|41.2|54.7|86.5|199.6|
> |+Naive CBVA|17.4|41.5|54.9|86.5|200.3|
> |+OP‑ToMe|17.5|41.9||86.9|201.7|
> |+Adaptive CBVA|17.7|42.0|55.6|86.8|202.1|
>
> # Scenarios token merging or adaptive CBVA struggle?
>
> One potential failure mode we have already mitigated is the brittleness of bipartite token‑merging since CLIP visual similarity does not accurately represent the degree of semantic overlap (fails to detect fine-grained action changes occurring within the same background).
> As stated in L214, we avoid per‑video threshold tuning by approximating the context count with a small, pre‑defined set of coarse clip‑count levels.
> This design choice not only (1) eliminates the expensive search for an exact context count for every video, but also (2) makes our method far less sensitive to noisy or ambiguous pairwise similarities.
> Simply put, our design in defining a discrete set of clip levels mitigates the sensitivity to specific similarity thresholding hyperparam.
> We will emphasise this rationale in the revised paper.
> As future work, we plan to incorporate temporal action cues and motion‑aware features into the merging criterion so that subtle action transitions can be recognized.
>
> # Does TCPL Amplify CLIP’s Semantic Errors?
>
> We agree that CLIP exhibits a high error rate for certain fine-grained concepts, such as directional cues or specific actions.
> Yet, in many practical scenarios (including PRVR) where coarse-grained semantic understanding is essential, the benefits of maintaining alignment with vision-language embedding spaces often outweigh the drawbacks stemming from their limitations.
> Indeed, we only use CLIP's knowledge to prevent overfitting; TCPL simply regularizes the student so it does not drift too far from CLIP’s global structure while retrieval losses and CBVA adapt the embeddings to PRVR.
> Consequently, any fine‑grained mistakes inherited from CLIP are free to be corrected during fine‑tuning, while TCPL preserves the robustness that CLIP’s coarse semantics provide.
>
> Lastly, while CLIP has limitations in fine-grained understanding, this is an active area of research [A], and we expect such issues to diminish as more capable models are developed (our method is backbone‑agnostic, so any stronger model can be integrated with minimal effort).
> Thus, we believe this concern, while valid, falls outside the scope of our work.
>
> [A] FG-CLIP: Fine-Grained Visual and Textual Alignment. ICML 2025.
>
> # In Eq. 4,5, why [EOS] for distillation?
>
> [EOS] is used for distillation for 3 reasons.
>
> 1. Global Summary:
> [EOS] token in CLIP text encoder serves as a global summary of the input.
> Since PRVR is typically performed at sentence level, we use [EOS] to capture the overall semantics of each query.
> Also, [EOS] is shown to convey more information than other tokens in CLIP [B].
>
> [B] Towards Understanding the Working Mechanism of Text-to-Image Diffusion Model. NeurIPS 2024.
>
> 2. Efficiency:
> Distilling only from [EOS] reduces computational overhead compared to token-wise distillation, enabling more efficient training.
>
> 3. T-V Alignment:
> In CLIP, text and visual features are aligned in a shared space through global features.
> Thus, we used [EOS] to follow this convention for stabilizing the training.
>
> # At which stage OP-ToMe is applied?
> We apologize for ambiguity.
> OP-ToMe is applied immediately after extracting video features from CLIP, and before passing them into any learnable modules.
> We will state this in the revised manuscript.

---

> > ### Author Response · Authors · 2025-08-06
> >
> > Dear Reviewer aktj
> >
> > Thank you for the thorough feedback.
> >
> > In our responses, we clarified **(1)** our core contributions, **(2)** Robustness to hyperparameter, **(3)** Efficiency and clarification about our Baseline, **(4)** Reproduction of missing baselines, **(5)** Reason for conducting ablation on QVHighlights along with an additional ablation study on ActivityNet Captions, **(6)** Hard Scenarios where our method struggles, **(7)** Our thought about CLIP’s semantic errors, **(8)** Reason for using text [EOS] token for distillation, and **(9)** When OP-ToMe is applied.
> >
> > Please let us know if any of these points need further elaboration.
> >
> > We are happy to provide additional details or experiments.
> >
> > Thank you again for your constructive review and time.
> >
> > Authors

---

> > ### Comment · Reviewer_aktj · 2025-08-06
> >
> > I appreciate the authors' rebuttal and the effort they put into the revision. However, several concerns remain that prevent me from recommending the paper for publication at this time. I encourage the authors to address these points in future work.
> >
> > 1. The paper lacks clarity regarding the dataset splits. For several datasets, the division into training, validation, and testing sets is not explicitly defined. This makes it difficult to understand how hyperparameters were tuned and raises questions about potential data leakage or overfitting to the test set.
> >
> > 2. The proposed method significantly increases the number of hyperparameters and loss functions compared to the baselines. While the authors state that their method is not heavily relying on hyperparameter tuning, the modification of certain parameters and the introduction of dataset-specific values (e.g. $\tau$) suggest otherwise. The authors only report ablation studies of hyperparameters on the QVHighlights dataset, it unclear whether the method is similarly sensitive to hyperparameters on the other datasets.
> >
> > 3. The rebuttal did not provide a direct analysis or experimental results on how CLIP's failure rate impacts the performance of the proposed method in the context of the PRVR task. While the authors cite related work, a task-specific discussion is necessary to fully address this concern.
> >
> > 4. The paper's description of its core methodology is unclear. The fundamental aspects of the PRVR task, such as how similarity scores are computed and how retrieval is performed, are not adequately explained. Furthermore, the implementation details of the proposed OP-ToMe are not sufficiently described, which makes it challenging for readers to understand and replicate the work. I agree with other reviewers that the paper overcomplicates several concepts while lacking fundamental explanations.
> >
> > 5. The core contributions of the paper appear to be incremental. The proposed TCPL can be seen as a form of knowledge distillation, and OP-ToMe is a minor variation of an existing method (ToMe). The concept of "Semantic Collapse" is also presented in a complex manner, when it could be more clearly framed as a known "one-to-many" or "many-to-many" problem.

---

> ### Author Response · Authors · 2025-08-07
>
> We sincerely appreciate the reviewer’s careful reconsideration of our rebuttal and additional feedback.
> We believe we can further clarify the remaining concerns.
>
> # Dataset Splits
>
> We respectfully clarify that the absence of explicit validation sets in PRVR datasets is a characteristic of established benchmarks (QVHighlights, ActivityNet-Captions, TVR, Charades-STA).
> To ensure fair comparisons, we strictly adhered to standard splits provided by benchmark creators.
> Critically, we used a unified hyperparameter configuration for all datasets (with $\tau$ being the only exception, adjusted slightly for each dataset’s distribution).
> By refraining from any dataset-specific hyperparameter tuning, we mitigated the risk of data leakage or overfitting.
> Indeed, compared to previous works such as GMMFormer (hyperparams listed in Tab. 8 in Appendix) and GMMFormer-v2 (listed in Tab. 7 in Appendix) which employ significantly different hyperparameters for each dataset, our method uses a single set of hyperparameters yet achieves state-of-the-art performance by a substantial margin.
> This demonstrates that our approach’s strong results are due to robustness and generalizability rather than overfitting.
>
>
> # Hyperparameter Sensitivity
>
> Regarding hyperparameter tuning concerns, we acknowledge that the similarity threshold ($\tau$) differs slightly across datasets.
> However, this variation naturally arises due to inherent differences in the distribution of video contexts per dataset (also can be inferred from ProtoPRVR's appendix [A] that statistics in data similarity distribution vary significantly).
> To empirically demonstrate our approach’s robustness to $\tau$, we conducted additional experiments on TVR, ActivityNet, and Charades.
> As shown below, varying $\tau$ causes minimal fluctuation in performance on each dataset (in the tables, * indicates the value used in our reported results).
> Thus, our model is evidently insensitive to minor changes in this hyperparameter.
> We will document these sensitivity analyses thoroughly in the revised Appendix to reinforce this point.
>
> [A] Prototypes are Balanced Units for Efficient and Effective Partially Relevant Video Retrieval
>
> |TVR $\tau$|R@1|R@5|R@10|R@100|SumR|
> |--|--|--|--|--|--|
> |0.70| 35.6 | 61.0 | 70.8 | 95.0| 262.4|
> |0.75| 35.5 | 61.2 | 71.1 | 94.9| 262.6|
> |0.80 *| 35.1 | 61.6 | 71.5 | 94.9| 263.1|
> |0.85| 35.1 | 61.2 | 71.2 | 95.0| 262.5|
> |0.90| 35.1 | 61.1 | 71.1 | 94.9| 262.2|
>
> |ANET $\tau$|R@1|R@5|R@10|R@100|SumR|
> |--|--|--|--|--|--|
> |0.70| 17.6 | 41.9 | 55.4 | 86.8| 201.7|
> |0.75| 17.8 | 41.9 | 55.4 | 86.7| 201.8|
> |0.80 *| 17.7 | 42.0 | 55.6 | 86.8| 202.1|
> |0.85| 17.7 | 42.1 | 55.3 | 86.8| 201.9|
> |0.90| 17.2 | 41.9 | 55.5 | 86.8| 201.4|
>
> |CHA $\tau$|R@1|R@5|R@10|R@100|SumR|
> |--|--|--|--|--|--|
> |0.70| 3.3 | 11.6 | 19.8 | 63.9| 98.6 |
> |0.75| 3.4 | 12.7 | 19.4 | 64.8| 100.3|
> |0.80| 3.4 | 12.0 | 18.7 | 64.5| 98.6 |
> |0.85*| 3.2 | 12.6 | 20.1 | 63.8| 99.7 |
> |0.90| 3.3 | 12.4 | 19.1 | 64.0| 98.9 |
>
> Finally, due to time constraints, we were unable to run a full hyperparameter sweep with the final model.
> However, we would like to share additional results based on the configuration used in Table 1(c), specifically the model combining (Baseline + TCPL + Naive CBVA) on ActivityNet.
>
> As previously mentioned in our rebuttal response to the question "Why are ablations only on QVH?", this configuration achieved a **SumR of 200.3** on ActivityNet.
>
> Below, we provide a broader hyperparameter analysis conducted on the ActivityNet dataset. These results demonstrate that the model remains largely stable across different hyperparameter choices, confirming its robustness. Moreover, they suggest that further dataset-specific tuning has the potential to yield even higher performance.
>
> |BASE+TCPL+NaiveCBVA on ANET|$\lambda^{CBVA}$:0.1|$\lambda^{CBVA}$:0.15|$\lambda^{CBVA}$:0.2|
> |--|--|--|--|
> |$\lambda^E,\lambda^A$:10,20|200.1|200.1|200.5|
> |$\lambda^E,\lambda^A$:15,30|**200.3**|200.9|199.8|
> |$\lambda^E,\lambda^A$:20,40|200.6|200.5|200.8|
>
> These findings further reinforce that our method is not overly sensitive to hyperparameter variation and remains reliable across a range of reasonable settings.

---

> ### Author Response · Authors · 2025-08-07
>
> # Methodological Clarity (PRVR & OP-ToMe)
>
> We respectfully emphasize the clarity in Sec.3.1 (for similarity matching) affirmed by other reviewers: Reviewer jkfJ stated, "Section 3.1 is well-written" and Reviewer fskT highlighted, "The manuscript is well-organized".
> Also, the similarity-based retrieval mechanism we use is a standard convention in the video retrieval literature and should be familiar to the community.
>
> Regarding OP-ToMe, we explicitly detailed the execution stage in our rebuttal (immediately following CLIP feature extraction, prior to learnable modules), and Fig.2 in the main manuscript clearly illustrates this.
> We thus expect no ambiguity about how and when OP-ToMe is applied.
>
> Additionally, we provided pseudocode for OP-ToMe in the rebuttal and will include it in the Appendix to ensure reproducibility (Indeed, Reviewer fskT mentioned that our algorithm for OP-ToMe clearly addressed this concern).
> Moreover, our source code has been submitted in the supplementary material, allowing the approach to be replicated precisely.
> Nonetheless, we are fully prepared to clarify any remaining uncertainties.
> If any aspect of our methodology is still unclear, we will gladly refine our explanations further in the revision.
>
> # Incrementality & Contribution Novelty
>
> While we understand that the perception of novelty can be subjective, we firmly believe our contributions go well beyond an incremental improvement.
> We explicitly identified and rigorously quantified the issue of semantic collapse in PRVR, which is a subtle yet fundamental challenge that was previously underexplored.
>
> Notably, Reviewer jkfJ observed that "the paper makes a valuable observation about current partially relevant video retrieval methods," acknowledging our clear and novel problem formulation. Similarly, Reviewer 3bLp stated that "the proposed method is novel with clear metrics and diagnostic experiments," reinforcing the originality and thoroughness of our work.
>
> Our contributions distinctly include:
>
> (1) Novel problem formulation: Identifying and addressing semantic collapse comprehensively across both text and video modalities.
>
> (2) Effective new methods: Our TCPL effectively distills global relational structures from CLIP to alleviate text semantic collapse.
> In contrast, prior methods simplistically push all same-video queries apart without maintaining a meaningful embedding structure.
>
> (3) Application-specific adaptation of ToMe: OP-ToMe ensures temporal coherence and adaptivity for untrimmed videos. Far from a trivial variant, OP-ToMe is a crucial adaptation for PRVR’s untrimmed video scenario. Furthermore, we leverage ToMe to construct target clips for cross-branch video alignment, with a novel design that dynamically assesses the number of valid contexts in each video, addressing domain-specific temporal constraints and dynamics.
>
> (4) Substantial empirical advances: Our approach achieves unprecedented improvements over previous SOTA methods (e.g., +8 SumR on QVHighlights, +5.2 on ActivityNet, +3.0 on TVR, +3.7 on Charades), demonstrating the practical significance of our contributions.
>
> We respectfully argue that these contributions substantially advance the understanding and performance of PRVR beyond mere incremental changes.
> We sincerely appreciate the reviewers’ constructive suggestions and remain committed to clarifying any lingering uncertainties.
> We hope that the additional analyses and clarifications above adequately address all remaining concerns, and we kindly request a reconsideration of our work in light of these detailed responses.

---

> ### Author Response · Authors · 2025-08-07
>
> # Impact of CLIP’s Failure Rate on TCPL
>
> We acknowledge the reviewer’s valid concern regarding potential amplification of CLIP’s semantic errors and agree that a more direct analysis of how CLIP's failure impacts our model's performance in the context of the PRVR task is necessary.
>
> To provide a more concrete answer, we share the results in a follow-up official comment.
>
> In the meantime, we offer a supplementary explanation focusing on the design advantages and specific mechanics of our methodology.
> The key principle behind TCPL is correction, not reliance, on CLIP.
> In our model, the primary learning signal is the PRVR retrieval loss, which plays the dominant role in aligning the model with the domain-specific semantics of both video and text.
> In practice, this means any inaccurate or overly coarse knowledge from CLIP is corrected through the retrieval training.
> TCPL serves to preserve only CLIP’s robust high-level semantics, thereby enhancing the model’s stability and generalization without locking in CLIP’s finer-grained errors.
> TCPL’s contribution is indeed significant. For example, as shown in Table 1 of our manuscript, incorporating TCPL alone improves SumR by 5.1 on QVHighlights. This gain remains stable even when we greatly increase the TCPL loss coefficients ($\lambda^E$, $\lambda^A$) up to (20, 40).
> Such stability suggests that TCPL is not simply carrying over CLIP’s semantic biases; instead, it complements the retrieval objective to boost performance.
>
> Furthermore, CLIP models are continuously improving in fine-grained understanding (for instance, FG-CLIP, ICML 2025), which means any semantic limitations of the current CLIP will be reduced in newer versions. Importantly, our approach is backbone-agnostic, so it can readily take advantage of future, more accurate vision-language models without modification.
>
> Your insightful feedback has been invaluable in improving the quality of our work.
> Thank you once again.

---

> ### Author Response · Authors · 2025-08-08
>
> # Impact of CLIP’s Failure Rate on TCPL - 2. Experiment Results
>
> Below, we provide an analysis to directly address the impact of CLIP’s failure on our method.
> As CLIP is widely known to struggle with fine-grained or context-rich semantics, we conduct this study on the TVR dataset.
> This is because most text queries in TVR involve multiple named entities or sequential actions that require the capability to comprehend complex temporal and contextual cues.
> To investigate how our model handles these limitations, we conducted an in-depth comparison between our retrieval model and CLIP’s zero-shot retrieval performance across all 10,895 queries in TVR.
>
> Specifically, for each query, we determined whether the ground-truth video was correctly retrieved within the top-$k$ results ($k \in \{1,10\}$) by ours and the CLIP zero-shot model.
> This allows us to construct a $2 \times 2$ confusion matrix shown below.
>
> Simply put, the matrix can be interpreted as:
>
> - Top-left: Both correct
> - Top-right: Ours correct / CLIP wrong
> - Bottom-left: Ours wrong / CLIP correct
> - Bottom-right: Both wrong
>
> **Recall@1 Confusion Matrix**
>
> |Ours \ CLIP Zeroshot (R@1) | Correct | Wrong |
> | --- | --- | --- |
> |**Correct**| 1277 | 2551 |
> |**Wrong**| 500 | 6567 |
>
> **Recall@10 Confusion Matrix**
>
> |Ours \ CLIP Zeroshot (R@10) | Correct | Wrong |
> | --- | --- | --- |
> |**Correct**| 4162 | 3627 |
> |**Wrong**| 386 | 2720 |
>
> This analysis reveals several key insights:
>
> - **Ours Corrects CLIP's Failures**: The most critical finding is in the top-right quadrant. At R@1, Ours correctly retrieves 2,551 queries that the zero-shot CLIP model gets wrong. This demonstrates that our method is not a passive recipient of CLIP's representations but actively refines them to handle complex queries. The improvement is even more pronounced at R@10, where ours rectifies 3,627 of CLIP's failures.
> - **Ours Retains CLIP's Strengths**: Ours successfully retrieves a large portion of the queries that CLIP also gets right (1,277 at R@1 and 4,162 at R@10), showing it preserves CLIP's effective coarse-level alignment.
> - **Minimal Negative Interference**: The number of cases where CLIP succeeds while Ours fails (bottom-left quadrant) is significantly smaller in comparison (500 at R@1, 386 at R@10). As our error analysis below shows, these "failures" are far less severe.
>
> This clearly shows that our method refines and adapts CLIP's embedding space to better suit domain-specific retrieval, without inheriting its failure modes.
>
> ---
>
> ## Error Case Analysis
>
> We further analyzed the queries corresponding to the off-diagonal cells of the R@1 matrix to understand the nature of the disagreements.
>
> ### **1. When CLIP fails and our model succeeds (2551 text queries)**
>
> These queries consistently involve complex temporal dynamics, multi-entity interactions, and contextual understanding that challenge CLIP's static image-text pre-training.
>
> Example text queries are:
>
> - *“Sebastian grabs his folder and stands up from the table” (ours rank: 1, CLIP rank: 237)*
> - *“Leonard shakes Leslie's hand and walks out of the room” (ours rank: 1, CLIP rank: 257)*
> - *“George pulls back on Meredith's rolling chair and drags her” (ours rank: 1, CLIP rank: 418)*
>
> Across these 2,551 queries where CLIP fails, the average rank from CLIP was **56**, indicating a fundamental misunderstanding of the required semantics.
>
> ### **2. When our model fails and CLIP succeeds (500 text queries)**
>
> These queries are typically simple and object-centric, requiring less compositional or temporal reasoning.
>
> For example:
>
> - *“House takes a sip of soda that is from the bottle” (CLIP rank: 1, ours rank: 2)*
> - *“Penny sends a text using the phone in her hands” (CLIP rank: 1, ours rank: 3)*
> - “Joey is folding his coat in the kitchen” *(CLIP rank: 1, ours rank: 2)*
>
> Crucially, even when ours fails in these cases, it is not by a large margin. The average rank for the ground-truth video across these 500 queries was merely **6.7**.
>
> ---
>
> ### Summary & Conclusion
>
> - When Ours corrects a CLIP failure, the performance gain is substantial (CLIP avg. rank $\approx$ 56 $\rightarrow$ ours rank 1).
> - When CLIP succeeds and Ours fails, the performance drop is minor (CLIP rank 1 $\rightarrow$ ours avg. rank $\approx$ 6.7)
>
> This evidence strongly supports our claim: TCPL does not inherit CLIP's failure patterns for complex, fine-grained retrieval.
> Instead, our retrieval-focused training objective effectively leverages CLIP's foundational semantic space while specifically targeting and correcting its weaknesses in temporal and compositional understanding.
> Therefore, our method's performance is not limited by CLIP's failure rate but is enhanced by a training process designed to overcome it for the specific demands of the PRVR task.

---

### Official Review · Reviewer_fskT · 2025-07-06

**Clarity:** 2
**Significance:** 3
**Originality:** 3
**Rating:** 4
**Confidence:** 4

**Summary:**

A regularization method is proposed to mitigate the semantic collapse issue in partial video retrieval. Compared with the previous methods, the author propose to mitigate the issue in both the text domain and vision domain. Specifically, in text domain, Text Correlation Preservation Learning (TCPL) is proposed. The text semantic relationship among text queries in a batch is preserved by maintaining the distances among the queries in CLIP feature space. In vision domain, Order-Preserving Token Merging (OP-ToMe) is proposed. The visual token is iteratively merged by refering the cosine similarities between each adjacent frames until pre-defined number of clips is achieved. Further, Cross-Branch Video Alignment is proposed to force the frame-level visual feature and clip-level visual feature to align with each other. The proposed method is evaluated on four benchmark text-to-video retrieval datasets to show its effectiveness.

**Questions:**

1.	For the semantic collapse issue, can directly aligning the text queries with their corresponding annotated video segment also help mitigate the issue?
2.	Why not using text queries relationship in the feature space constructed by a pure NLP encoder as the training supervision guidance?

**Ethical Concerns:**

["NO or VERY MINOR ethics concerns only"]

**Final Justification:**

Considering the good performance and the insights the authors share with the community, I tend to rate borderline accept. And I suggest the authors to repolish the paper to make it easy to understand.

**Limitations:**

Yes.

**Paper Formatting Concerns:**

No concern,

**Quality:**

3

**Strengths And Weaknesses:**

**Strength:**

1.	The proposed method is extensively evaluated to show its effectiveness;

2.	The proposed method achieves the state-of-the-arts performance;

3.	The manuscript is well-organized;


**Weakness:**


1.	In TCPL, the training objective is to mimic the relationships among text queries constructed in the feature space of CLIP model. However, the sematic collapse issue may also occur in the CLIP feature space, considering that the training objective of CLIP is to align the text feature with the image feature. Why not using the feature space constructed by a pure NLP encoder?

2.	The description of the proposed method is not clear. For example, there is no equation or figures to illustrate how the OP-ToMe works, and it is not easy to re-implement it;

---

> ### Author Rebuttal · Authors · 2025-07-31
>
> # (TCPL) CLIP may also be vulnerable to semantic collapse. Why not distilled by NLP encoder?
>
> Semantic collapse in PRVR primarily arises from the deterministic learning with partially labeled data, where unannotated pairs are treated as negatives, even though they may share positive semantic contexts.
> While CLIP also employs contrastive learning over pairwise data, it is trained on a significantly larger and more diverse set of image-text pairs.
> This diversity helps mitigate the risk of semantic collapse, making CLIP more robust compared to models trained solely on narrow or task-specific datasets.
>
> Moreover, despite potential vulnerabilities, CLIP’s vision-language aligned space offers strong semantic unification across modalities [A], which is critical for PRVR, where text and video need to be embedded into a common space effectively.
> In contrast, using an NLP encoder for distillation may reduce the risk of collapse due to its inherently relational training, but it lacks alignment with the visual backbone.
> This misalignment limits its effectiveness for vision-language tasks like PRVR, as evidenced by the empirical performance gap shown in the table below.
> We observe the same performance patterns when a RoBERTa encoder is used as both the base text backbone and the TCPL source model.
> Experiments are conducted on ANET and CHA due to their readily available GT and fast training time.
>
> [A] Unified Visual Relationship Detection with Vision and Language Models. ICCV. 2023.
>
> ### AcitivityNet
> | Model($\lambda^E, \lambda^A$) | R1 | R5 | R10| R100 | SumR |
> |--|--|--|--|--|--|
> | **CLIP base** | **15.7** | **39.2** | **52.9** | **85.9** | **193.8** |
> | CLIP+ Rob TCPL (10,20) | 13.2 | 34.1 | 47.9 | 83.3 | 178.5 |
> | CLIP+ Rob TCPL (15,30) | 12.3 | 33.3 | 46.7 | 82.5 | 174.8 |
> | CLIP+ Rob TCPL (20,40) | 12.0 | 32.8 | 45.7 | 81.8 | 172.3 |
>
> ### Charades
> | Model($\lambda^E, \lambda^A$)  | R1 | R5 | R10| R100 | SumR |
> |--|--|--|--|--|--|
> | **CLIP base**| **2.6** | **11.2** | **18.3** | **62.6** | **94.8** |
> | CLIP+ Rob TCPL (10,20) | 3.4 | 11.2 | 18.2 | 62.4 | 95.1 |
> | CLIP+ Rob TCPL (15,30) | 3.1 | 10.6 | 17.4 | 61.6 | 92.7 |
> | CLIP+ Rob TCPL (20,40) | 3.0 | 11.0 | 17.6 | 60.1 | 91.7 |
>
> # Unclear description, especially OP-ToMe.
> We sincerely apologize that re-implementation based on our current description may be difficult.
> To provide a detailed description for OP-ToMe, we provide an algorithm below.
>
> ────────────────────────────────────────────────────────────────────────────
>
> Algorithm 1. Order‑Preserving Token Merging (OP‑ToMe)
> ```
> Input : Frame tokens  V_f ∈ ℝ^{B_v × L_f × d_v}
>         Merge rate    N %      (percentage of pairs to merge per iteration)
>         Iterations    M
> Output: Clip tokens   V_c ∈ ℝ^{B_v × L_c × d_v}   where L_c = 32
> ```
> ```
> 1.  Initialize token‑size vector  s ← 1_{L_f}        # each token covers 1 frame
>
> 2.  for m = 1 … M do
> 3.      # 1. Compute similarity of disjoint adjacent pairs (odd indices 1,3,5,…)
> 4.      for i ∈ {1, 3, 5, … , L_f‒1}:
> 5.          S[i] ← cos( V_f[i], V_f[i+1] )
> 6.      end for
> 7.
> 8.      # 2. Pick the top‑N% most‑similar adjacent pairs
> 9.      P ← indices of top‑N pairs in S
> 10.
> 11.     # 3. Merge each selected pair
> 12.     for (i, i+1) ∈ P (processed left→right):
> 13.         V_merge ← ( s[i]·V_f[i]  +  s[i+1]·V_f[i+1] ) / ( s[i] + s[i+1] )
> 14.         V_f[i]← V_merge                 # replace left token by the merge
> 15.         delete V_f[i+1]                   # remove right token
> 16.         s[i] ← s[i] + s[i+1]            # update size
> 17.         delete s[i+1]
> 18.     end for
> 19.
> 20.     L_f ← |V_f|                           # new token length
> 21.     if L_f ≤ 32 then break
> 22. end for
> 23.
> 24. return V_c ← V_f                          # 32 order‑preserving clip tokens
> ```
>
> In addition, to improve the readability of adaptive CBVA, we will rewrite the opening sentence of L214-218 as:
> "However, optimizing the number of semantics per video is costly during the token merging process.
> Therefore, we instead pre-define a discrete set of clip numbers based on a fixed merge rate, and then match each video to the level that best reflects its internal similarity structure (number of different semantics).
> To initially establish a discrete set of clip levels, we define $N\%$ to denote the merge rate and $C_\text{min}$ to represent the minimum number of semantically different clips in each video.
> Then, we generate $K$ levels of clip number candidates $\{L_c^i\}_{i=1}^{K}$ by recording clip number after each merge step as:", and also provide an algorithm for preparing adaptive lengths of clips for Adaptive CBVA (shown below).
> Specifically, the process of pre-computing a discrete set of different levels of clip number (number of semantics) is shown in Algorithm 2.
> Algorithm 3 shows the process of per-video merging for Adaptive CBVA.
>
> ────────────────────────────────────────────────────────────────────────────
>
> ### Algorithm 2. Pre-computing the discrete clip-count levels before Adaptive CBVA
> ```
> Input : Initial clip length   L_c^1 = L_c      (e.g., 32)
>         Merge‑rate            N %              (percentage applied per step)
>         Minimum clip count    C_min
> Output: Candidate list        L = [L_c^1 , L_c^2 , … , L_c^K]
> ```
> ```
> 1.  i ← 1                                   ;  L ← [ L_c^1 ]
> 2.  while  L_c^i > C_min  do
> 3.      # Compute next coarser clip length
> 4.      L_new ← max( 2 × floor( ( L_c^i − (L_c^i /2) × (N/100) + 1 ) / 2 ),
> 5.                     C_min )
> 6.      if  L_new = L_c^i   then  break      # no further reduction possible
> 7.      append  L_new  to  L
> 8.      i ← i + 1
> 9.  end while
> 10. K ← |L|                                  # number of discrete levels
> 11. return  L
> ```
> ────────────────────────────────────────────────────────────────────────────
>
> ### Algorithm 3. Constructing Merged Clips for Adaptive CBVA
> ```
> Input : Clip tokens        V_c  ∈ ℝ^{B_v × L_c × d_v}
>         Global candidate   L    = [L_c^1 … L_c^K]   # pre‑computed clip counts
>         Merge‑rate         N%                      # applied at each merge
>         Similarity thresh. τ
> Output: Adapted clip tokens  Ṽ_c   (length  L_c* )
> ```
> ```
> # Stage 1 – Estimate internal similarity
> 1.  S ← cosine‑similarity matrix of frozen V_c              # L_c × L_c
> 2.  ω ← |{(i,j) : S_ij > τ , i≠j}| / [L_c (L_c − 1)]        # high‑sim ratio
>
> # Stage 2 – Select merging depth k*
> 3.  if  ω ≤ 1 − 1/K                     then   k* ← 1        # keep all clips
> 4.  else
> 5.       k* ← min{ k ∈ {2 … K} |  ω > (K − k)/K }           # pick coarser level
> 6.  end if
>
> # Stage 3 – Merge clips (k* − 1 times)
> 7.  Ṽ_c ← V_c
> 8.  for  m = 1 … (k* − 1)  do
> 9.        Apply bipartite token‑merging (ToMe [B]) to Ṽ_c at rate N %
> 10. end for
> 11. L_c* ← |Ṽ_c|
> 12. return  Ṽ_c
> ```
>
> [A] Token Merging: Your ViT But Faster. ICLR 2023.
>
> We claim that this operation (pre-computing levels of different clip numbers) offers two benefits over determining whether to merge the paired clips based on their similarity.
>
> (1) Efficiency:
> Clip count candidates are pre-computed once, and each video is matched to the most appropriate level, thereby no recursive per-pair merge decision is needed.
> By contrast, deciding at every pair whether to merge would insert an extra similarity-threshold branch into each pairing step, increasing the computational overhead.
>
> (2) Robustness:
> Visual similarity is not always accurate in measuring the contextual similarity in text descriptions.
> While thresholding the pairwise similarity introduces brittle sensitivity to the chosen threshold value, we claim that a coarsely discretised set of clip count candidates alleviates such sensitivity and yields more stable performance across diverse video domains.
>
> We will integrate this clarification, as well as the Algorithms in the appendix.
> Also, we will carefully proofread the entire paper to improve clarity.
>
> # Does the use of annotated video segments mitigate Semantic collapse?
> As mentioned by the reviewer, using the annotated video segment may partially mitigate the semantic collapse by encouraging the model to pull each query toward its exact visual span.
> However, this strategy (i) depends on dense temporal annotations that are costly and often impractical in real-world PRVR settings, (ii) still leaves unlabeled portions of the video collapsed together, and (iii) does not address semantic collapse among different videos that share similar events.
>
> To verify whether the annotated video segment can only partially mitigate the semantic collapse, we trained a model to align each query with its annotated span using the GT‑segment.
> We tested this model on ANET, CHA, and TVR datasets where GT timestamps are readily available.
> To illustrate, the use of GT timestamps results in a performance boost in ActivityNet-Captions (+4.3), but has little effect on TVR or Charades.
> Moreover, our framework boosts the baseline by +7.9 on ANET, surpassing the gains achieved with ground-truth segments and indicating that it tackles semantic collapse more effectively than simply injecting GT boundaries.
> Therefore, we argue that ground-truth timestamps alone are insufficient; additional labels that capture the semantic relationships between video segments are also necessary to effectively address semantic collapse.

---

> > ### Author Response · Authors · 2025-08-06
> >
> > Dear Reviewer fskT,
> >
> > Thank you very much for the constructive feedback, especially for highlighting both the strengths and the points that need clarification.
> >
> > Our responses include **(1)** Comparing CLIP and NLP encoder as the source model for distillation, **(2)** Clarification for our methods, and **(3)** Experiment regarding whether annotated video segments address semantic collapse.
> >
> > We would be grateful to know whether these additions address your main concerns or if there are particular details you would like us to elaborate on further.
> >
> > Lastly, the new materials will be included in the appendix, and we will release code to facilitate future reproduction.
> > Thank you again for your valuable time and insights.
> >
> > Authors

---

> > > ### Comment · Reviewer_fskT · 2025-08-06
> > >
> > > Thanks for the rebuttal. Most of my concerns have been addressed, and I tend to retain my rating.

---

> ### Author Response · Authors · 2025-08-08
>
> Dear Reviewer fskT,
>
> We sincerely appreciate the time and effort you have dedicated to reviewing our submission and engaging with our rebuttal. We are grateful for the opportunity to address your concerns and provide further clarification on our work.
>
> Should any additional questions or points of discussion arise during the discussion phase, we would be pleased to offer further explanation.
>
> Thank you once again for your thoughtful feedback and your contributions to the review process.

---

### Note · Authors · 2025-08-12

# Dear Area Chair and Reviewers,

We sincerely thank you for the thorough and constructive feedback.
We are encouraged that discussions with Reviewers jkfJ, 3bLp, and fskT successfully addressed their concerns.

For your final consideration, we wish to summarize our responses to the key remaining points:

## A Novel Contribution to a Fundamental Problem

We rigorously formulated and quantified "semantic collapse" in PRVR.
Other reviewers characterized this as a "valuable observation" and our method as "novel with clear metrics".
Our consistent SOTA gains (up to +8 SumR on QVHighlights) demonstrate the impact of solving this core challenge.

## Our Model Corrects, Not Inherits, CLIP’s Failures

Our strongest evidence is the new 2x2 analysis on TVR.
At R@1, we correctly retrieve 2,551 queries that zero-shot CLIP failed (improving CLIP's avg. rank from $\approx$ 56 to our rank 1).
This empirically refutes concerns about inheriting CLIP's limitation and proves our method actively refines the embedding space.

## Robust and Reproducible Results

For the dataset split, we used official train/test splits (PRVR lacks a separate validation set).
Also, we claim that our method is not overfitted to the test set as we use a single hyperparameter setup across datasets except $\tau$.
Extensive experiments regarding the sensitivity of $\tau$ on TVR, ANET, and CHA are provided in the rebuttal response, confirming the robustness of our method (results on QVH are in the paper).
This new study in rebuttal shows the minimal fluctuation when varying $\tau$ on TVR (SumR: 262.2–263.1), ANet (201.4–202.1), and Charades (98.6–100.3).
Our code will be released to ensure reproducibility.

## Clarity

We provided step-by-step pseudocode for (i) OP-ToMe, (ii) Precomputing discrete clip-count levels, and (iii) Adaptive CBVA in our rebuttal.
These algorithms have addressed the other reviewers' concerns regarding clarity.

All our rebuttal responses, including these results, will be added to the manuscript / Appendix.
We believe these points clearly demonstrate our paper's novelty, robustness, and impact.

We thank you again for your time and careful consideration.

---

### Decision · Program_Chairs · 2025-09-17

**Decision:**

Accept (poster)

**Comment:**

This paper addresses the challenge of semantic collapse in Partially Relevant Video Retrieval (PRVR) and proposes a novel framework combining Text Correlation Preservation Learning and Cross-Branch Video Alignment (CBVA), enhanced with order-preserving token merging and adaptive CBVA. The approach is well-motivated and effectively mitigates embedding collapse, leading to substantial improvements in retrieval accuracy on PRVR benchmarks. The reviewers appreciated the clear problem formulation, strong empirical results, and practical relevance of the proposed solution. While some concerns were raised about limited ablation studies and generalizability analysis, these do not significantly undermine the overall contributions or validity of the findings. And the authors respond to these minor concerns in the discussion period. Given its novelty, strong experimental performance, and potential impact on text-video retrieval, I recommend acceptance.